

# Optimization of pour-over coffee extraction variables using reinforcement learning

Arif Bramantoro[1], Moch Riyadi Maskur A[2], Ahmad A. Alzahrani[3], Zhahirah Jeffery[1] and Ary Mazharuddin Shiddiqi[4]

[1] School of Computing and Informatics, Universiti Teknologi Brunei, Bandar Seri Begawan, Brunei
[2] Department of Computer Science, Budi Luhur University, Jakarta Selatan, DKI Jakarta, Indonesia
[3] Faculty of Computing and Information Technology, King Abdul Aziz University, Jeddah, Saudi Arabia
[4] Department of Informatics, Institut Teknologi Sepuluh Nopember, Surabaya, Indonesia

Corresponding authors
Arif Bramantoro,
arif.bramantoro@utb.edu.bn
Ary Mazharuddin Shiddiqi,
ary.shiddiqi@its.ac.id

## ABSTRACT

Pour-over coffee brewing is influenced by multiple interdependent variables—roast level, grind size, brew ratio, extraction time, water temperature, and total dissolved solids (TDS)—that collectively determine the final flavor and quality. This study explores the optimization of these variables using reinforcement learning (RL) and compares its performance with three common machine learning models: K-nearest neighbors (KNN), decision tree, and support vector machine (SVM). The RL agent was designed to balance exploration and exploitation, aiming to maximize rewards by adjusting brewing parameters. Data were gathered from both professional baristas and homebrewers to train and evaluate the models. The RL approach achieved the highest accuracy (90.00%), precision (90.76%), recall (90.00%), and F1-score (90.08%), outperforming KNN (accuracy: 88.33%, F1-score: 88.90%), and significantly surpassing decision tree and SVM classifiers, both of which exhibited high recall (100%) but only 50.00% accuracy and 66.67% F1-score. These findings highlight RL's superior capability for optimizing complex, interdependent variables in pour-over coffee brewing, offering a systematic and dynamic approach to improving consistency and quality.

## INTRODUCTION

The coffee industry has undergone significant transformations over the years, often described in terms of three distinct waves: the first, second, and third waves (*Schenker & Rothgeb, 2017*). The first wave focused on making coffee widely accessible as basic commodity, emphasizing mass production and convenience, typically through instant coffee. The second wave introduced the concept of specialty coffee and the coffee shop experience, with an emphasis on origin, branding, and espresso-based beverages. The third wave, which emerged in Indonesia around 2008, marked a shift in how coffee is perceived—from a simple beverage to a sophisticated drink that is highly valued and

meticulously brewed (*Gunawan et al., 2021*). Therefore, in the third wave, coffee brewing techniques have become highly diverse and carefully considered, positioning coffee not merely as an instant beverage but as a crafted and appreciated experience.

This shift has driven a steady increase in global coffee consumption, leading to the popularity of various brewing methods, including the pour-over technique. The pour-over method is a manual brewing process that involves filtering coffee through a specific filter, commonly a paper filter, using a filter holder or dripper. Popular drippers include the Hario V60, Kalita Wave, Chemex, and others. This method focuses on carefully controlling brewing variables to enhance the complexity and richness of coffee flavors (*Hidayat, Anugrah & Munir, 2019*; *Yu et al., 2021*). Key variables influencing the pour-over brewing process include extraction level, brew ratio, grind size, brew time, water temperature, agitation, and total dissolved solids (TDS, the concentration of dissolved substances in the brewed coffee) (*Santanatoglia et al., 2023*). The extraction level, determined by the interaction of these variables, plays a critical role in the final taste of the coffee. The brew ratio refers to the proportion of coffee grounds to water, while grind size affects particle surface area and extraction yield. A finer grind size increases the contact surface area, enhancing extraction but also impacting brew time. Similarly, water temperature influences solubility and viscosity, with higher temperatures increasing solubility while reducing viscosity, both of which affect extraction outcomes (*Schmieder et al., 2023*).

In addition to these internal brewing variables, external factors such as coffee type, post-harvest processing, and roast level must also be considered. While these factors cannot be adjusted during brewing, the process must adapt to their influence. For instance, roast levels affect the concentration of caffeic acid, a key compound that decreases with increased roasting due to chemical reactions. This compound is predominantly extracted at the early stages of brewing, with its concentration declining in subsequent pourings (*Sano et al., 2019*).

Among all brewing variables, the brew ratio is often considered the most critical. An incorrect coffee-to-water ratio leads to suboptimal extraction. Under-extracted coffee has underdeveloped flavors and a watery taste, while over-extraction results in a dominant bitterness and diminished flavor complexity (*Cordoba et al., 2020*). These outcomes highlight the importance of precise control over brewing parameters to achieve optimal extraction levels.

Given the complexity of the brewing process, preparing pour-over coffee requires significant knowledge and expertise. However, the advent of machine learning has introduced new opportunities for optimizing such processes. Machine learning techniques have been widely applied in food preparation, including the use of reinforcement learning (RL) for recipe optimization (*Fujita, Sato & Nobuhara, 2021*). RL is a branch of machine learning that uses a trial-and-error approach to map situations (states) to actions in order to maximize cumulative rewards. Think of RL as a barista learning to brew the perfect cup of coffee through trial and error, much like a chef perfecting a recipe in the kitchen. The barista might experiment with a new grind size or water temperature (exploration) to see if it improves the flavor, or stick to a known recipe that consistently produces a great cup

(exploitation). Over time, the barista learns which adjustments lead to the best coffee, guided by feedback—like a reward for a delicious brew.

Unlike traditional supervised learning, RL does not explicitly instruct the agent on which actions to take but allows it to learn through experimentation (*Bekkemoen, 2024*; *Ernst et al., 2024*; *Ris-Ala, 2023*; *Sutton & Barto, 2018*). RL has been successfully applied in various fields. For example, in education, RL improves personalized learning experiences by recommending adaptive learning paths based on feedback (*Amin et al., 2023*). In healthcare, RL has been employed to manage complex conditions like diabetes, enabling agents to optimize decision-making based on dynamic patient states (*Yau et al., 2023*). In finance, RL powers automated trading systems, helping agents determine optimal buy and sell actions for maximum profit (*Dharmawan & Bintang, 2020*).

The pour-over brewing process is dynamic and interdependent, with multiple variables—such as water temperature, brew ratio, brew time, and grind size—interacting to influence the final cup profile. This complexity makes the process well-suited for optimization using machine learning. Among the various techniques, RL is particularly advantageous due to its ability to learn optimal strategies through trial-and-error interactions within an environment, rather than relying solely on labeled datasets.

While other machine learning methods, such as supervised learning (*e.g.*, k-nearest neighbors (KNN)) or unsupervised learning, have been applied in coffee-related studies for tasks like classification of roast levels (*Anita & Albarda, 2020*), they are less suited for the dynamic optimization required in pour-over brewing. Supervised learning relies on labeled datasets, which are difficult to obtain for the vast combinations of brewing variables, and unsupervised learning lacks the ability to directly optimize for a reward signal like flavor quality. In contrast, RL excels in sequential decision-making and adapts to dynamic environments through trial-and-error, making it ideal for balancing interdependent brewing parameters to achieve the desired flavor profile.

In this study, we propose the use of RL to optimize pour-over coffee brewing. By employing an RL agent, we aim to identify the best actions for each brewing variable based on reward signals, leading to consistent, high-quality extractions. To evaluate the effectiveness of the proposed RL approach, KNN is used as a benchmark method. KNN is a simple yet effective supervised learning algorithm that classifies instances based on the majority label of their closest neighbors in the feature space. Its non-parametric nature and ease of implementation make it a common baseline in various machine learning tasks, including those involving sensor data or low-dimensional inputs. Although KNN does not account for sequential dependencies or dynamic interactions between variables—key characteristics of the pour-over brewing process—it provides a useful point of comparison for assessing the advantages of RL, particularly in terms of adaptability and consistency in complex, real-world conditions.

Recent academic work also reflects a growing interest in the application of artificial intelligence for coffee and food preparation. *Motta et al. (2024)* provided a comprehensive review of machine learning techniques used for coffee classification tasks, such as predicting roast levels, bean origin, and flavor attributes. Similarly, *Kim & Kim (2024)* applied convolutional neural networks (CNNs) and robotic automation to optimize food

preparation, showing how AI can be used to tailor processing parameters to desired outcomes. These studies highlight the emerging role of AI in personalizing and enhancing beverage quality. Our work builds on this direction by applying reinforcement learning not to classify or replicate, but to dynamically optimize the sequential brewing decisions in pour-over coffee, with the goal of producing consistent, high-quality extractions across diverse conditions.

## COFFEE EXTRACTION

Pour-over coffee is a popular brewing technique that filters coffee through a dripper equipped with a paper filter. This method involves pouring hot water from a kettle over coffee grounds held in the filter, extracting the flavors and aromas of the coffee (*Hidayat, Anugrah & Munir, 2019*). At its core, coffee extraction is a critical step in brewing, involving the absorption of water by coffee grounds, the dissolution of soluble compounds, and the separation of the resulting brew from the grounds. This mass transfer process, where hot water interacts with coffee particles, directly determines the flavor and quality of the final cup (*Sano et al., 2019*). Despite its brevity, typically lasting only a few minutes, coffee extraction demands precision to achieve the desired taste profile (*Cordoba et al., 2020*).

The effectiveness of coffee extraction is often measured by the extraction percentage, which represents the proportion of soluble compounds removed from the grounds. While preferences for extraction levels can vary, the Specialty Coffee Association (SCA) recommends an ideal extraction range of 18% to 22%, paired with TDS concentrations between 1.15% and 1.55% (*Lingle, 2011*). Under-extracted coffee, caused by insufficient dissolution, tends to be sour and lacks complexity, while over-extracted coffee, resulting from excessive dissolution, can taste overly bitter and harsh.

Several variables influence coffee extraction, requiring careful control to achieve optimal results. These include roast level, grind size, brew ratio, brewing time, and water temperature. Roast level determines the degree to which coffee beans are heated, which affects their chemical composition, flavor profile, and appearance. During roasting, reactions like caramelization and the Maillard process produce the compounds that contribute to coffee's distinctive flavors and aromas (*Hakim, Djatna & Yuliasih, 2020*). Lightly roasted coffee often retains floral and acidic notes, while darker roasts emphasize bitterness and body, often at the expense of subtle flavors.

Grind size, which dictates the surface area of coffee particles, also plays a key role. Finer grinds increase the surface area, speeding up extraction, while coarser grinds reduce it, slowing the process. Adjusting grind size is essential to balance the extraction rate. Similarly, the brew ratio—the proportion of coffee grounds to water—affects the concentration and intensity of the resulting brew. Adjustments to the brew ratio often depend on other factors, such as grind size and desired strength.

Brew time and water temperature are equally important. Longer brewing times generally allow for greater extraction but risk over-extraction if not carefully managed. Meanwhile, water temperature affects the solubility of coffee compounds, with higher

temperatures accelerating extraction. Maintaining the correct balance of these variables ensures a well-rounded flavor profile.

The pour-over brewing method involves several key steps, each of which contributes to the final cup's quality. Preparation begins with assembling the necessary tools, such as a dripper, filter, gooseneck kettle, coffee server, thermometer, scale, and timer. Flushing the paper filter with hot water helps remove any residual paper flavor, ensuring that it does not interfere with the coffee's taste. The blooming phase follows, where a small amount of water is poured over the grounds to release carbon dioxide, enabling more even extraction. This step is crucial, typically lasting 30–45 s.

During the pouring phase, water is added in a slow, spiral motion to ensure even saturation. The flow rate and technique directly impact extraction uniformity. Once the brewing process is complete, the coffee is ready to be enjoyed, showcasing the interplay of precise techniques and variable adjustments (*Dashwood, 2017*; *Stephenson, 2019*).

While foundational texts such as (*Sutton & Barto, 2018*) have laid the groundwork for RL, recent literature has expanded and updated these principles. *Chadi & Mousannif (2023)* provide a comprehensive review of RL algorithms, tracing their evolution from basic Q-learning to advanced methods like Proximal Policy Optimization. Similarly, *Kommey et al. (2024)* offer an extensive analysis of RL's historical development, core challenges, and current advancements, highlighting the field's rapid progression and diversification.

The decision-making process in RL can be modeled using a Markov decision process (MDP), a framework for sequential decision-making in dynamic environments. MDPs define states, actions, transition probabilities, rewards, and discount factors. States represent the environment's current conditions, while actions denote the possible choices. Transition probabilities describe the likelihood of moving from one state to another, and rewards measure the value of outcomes. Discount factors balance immediate and future rewards (*Puterman, 2014*). A central challenge in RL is managing the balance between exploration and exploitation. Exploration involves testing unfamiliar actions to gather information, while exploitation leverages existing knowledge to maximize rewards. Achieving this balance is essential for effective learning and decision-making (*Bokade, Jin & Amato, 2023*).

In the context of coffee extraction, RL agents can model brewing as a dynamic system, systematically testing combinations of variables to optimize outcomes. This has practical implications for improving consistency, reducing waste, and enhancing the sensory qualities of brewed coffee. By leveraging data-driven insights, RL could revolutionize how coffee is brewed, benefiting professionals and enthusiasts alike.

## METHODS

This study is a quantitative, experimental research that quantifies coffee extraction levels based on the actions and rewards obtained by an agent. The research employs reinforcement learning to evaluate the complexity of brewing variables and their impact on extraction efficiency in pour-over coffee. This approach enables a systematic assessment of process parameters, which is essential for optimizing brewing performance.

The sampling method used in this study is non-random, specifically convenience sampling, in which participants are selected based on their availability and ease of access. The research sample consists of baristas and home brewers who regularly prepare pour-over coffee, ensuring that the data reflect practical and experienced insights. Therefore, no dataset has been curated or uploaded by an external source. This method was chosen to reflect realistic brewing practices by experienced individuals while working within practical time and resource constraints typical in pilot studies. Although convenience sampling may limit generalizability, it is appropriate for early-stage experimental validation, where feasibility and relevance to actual users are prioritized.

Data were collected from both baristas and home brewers at Ujala Café & Roastery in Cianjur, Indonesia. The primary goal of data collection was not for classification or clustering purposes but to support the agent in selecting and evaluating its actions effectively. Additional observations were made regarding the consistency of extraction levels, which contributed to refining the reinforcement learning model.

To support reproducibility, this study follows best practices in machine learning research. All code and simulation environments used are publicly available at https://github.com/ArifBramantoro/coffee-rl-optimization under an MIT license. Experiments were conducted using Python 3.9 with Stable-Baselines3 (v1.8), NumPy (v1.24), and Gym (v0.26). Random seeds were fixed for environment resets and model initialization to ensure consistent results. Key hyperparameters, including buffer size, learning rate, and discount factor, are explicitly detailed in the Methods section. The experiments were run on a MacBook M2 with 256GB SSD and 8GB RAM, without GPU acceleration. The complete dataset of 3,600 simulated brewing sequences is openly available at https://dx.doi.org/10.6084/m9.figshare.28503464.

The ingredients used for brewing pour-over coffee in this study were carefully selected. The coffee was sourced as a Single Origin Gayo variety with a natural post-harvest process and a medium roasting level. Three grind sizes were considered: fine to medium (600 μm), medium (900 μm), and medium to coarse (1,200 μm). Mineral water with a TDS level of 2 parts per million was used to maintain consistency in the brewing process.

The brewing equipment included a Hario V60 Electric Coffee Grinder, a Hario V60 Dripper, and Hario Paper Filters. A gooseneck kettle with a thermometer was used to control water temperature during brewing. Additional tools included a Hario scale with a timer for precise measurements and a coffee server for collecting the brewed coffee.

## Data preprocessing steps

To collect meaningful insights, brewing data were sourced from both baristas and homebrewers experienced in preparing pour-over coffee. For medium roast coffee, 10 data points were collected exclusively from baristas as shown in Table 1. These data were analyzed to identify the most influential variables for optimizing coffee extraction. The variables considered included grind size, brew ratio, brew time, and temperature.

The first step in data preprocessing involved coding and categorization. Roasting levels and grind sizes were categorized as shown in Tables 2 and 3, respectively. For the brew ratio, the ratio of coffee to water was used, such as 1:15, where 1 gram of coffee corresponds

**Table 1 Brewing data for medium roasting level.**

| Barista data | | | | Level roasting: light-medium | | | |
|---|---|---|---|---|---|---|---|
| Name | Gender | Age | Experience | Grind size | Brew ratio | Brew time | (°C) |
| 1st Barista | Male | 22 | >1 year | Medium-coarse | 1:16 | 3 m 30 s | 90 |
| 2nd Barista | Male | 24 | 4 months | Fine-medium | 1:12 | 2 m | 86 |
| 3rd Barista | Male | 19 | <1 year | Medium | 1:16 | 3 m | 90 |
| 4th Barista | Male | 27 | >3 years | Fine-medium | 1:13 | 2 m 25 s | 87 |
| 5th Barista | Male | 35 | 10 years | Medium-coarse | 1:16 | 3 m 40 s | 91 |
| 1st Homebrewer | Female | 28 | >3 years | Medium | 1:15 | 2 m 45 s | 89 |
| 2nd Homebrewer | Male | 26 | >3 years | Medium-coarse | 1:18 | 4 m | 92 |
| 3rd Homebrewer | Male | 24 | <1 year | Fine-medium | 1:14 | 3 m | 88 |
| 4th Homebrewer | Female | 21 | >2 years | Medium-coarse | 1:17 | 3 m 50 s | 92 |
| 5th Homebrewer | Female | 22 | >2 years | Medium | 1:14 | 2 m 30 s | 88 |

**Table 2 The grind size levels ranging from 1 to 7, where each numerical level corresponds to a specific coffee grind texture classification.**

| Code | Grind size level |
|---|---|
| 1 | Extra fine |
| 2 | Fine |
| 3 | Fine to medium |
| 4 | Medium |
| 5 | Medium to coarse |
| 6 | Coarse |
| 7 | Extra coarse |

**Table 3 Low and high values for medium roasting.**

| Level roasting: medium | | | | | | |
|---|---|---|---|---|---|---|
| Grind size | Brew ratio | | Brew time | | Temperature (°C) | |
| | Min | Max | Min | Max | Min | Max |
| Fine-medium | 1:12 | 1:14 | 2 m | 3 m | 86 | 88 |
| Medium | 1:14 | 1:16 | 2 m 30 s | 3 m | 88 | 90 |
| Medium-coarse | 1:16 | 1:18 | 3 m 30 s | 4 m | 90 | 92 |

to 15 ml of water. Brew time was measured in minutes and seconds, while temperature was recorded in degrees Celsius. The low and high values for each variable were determined from the minimum and maximum observed in the collected data.

The second step focused on normalizing the variables. Each variable (grind size, brew ratio, brew time, and temperature) was scaled to ensure compatibility with the reinforcement learning framework. The low and high values for each variable were

determined from the minimum and maximum observed in the collected data. In cases where data points were incomplete or missing, interpolation was used to estimate values based on the nearest available data. From the baristas' brewing data, three grind sizes were commonly identified for the medium roasting level. Each grind size had a unique range for brew ratio, brew time, and temperature. Similarly, for the medium to dark roasting level, three grind sizes were applicable, with corresponding ranges outlined in Table 3. These classifications and ranges provided a structured basis for further experimentation and analysis.

The third step involved splitting the dataset for KNN modeling. The dataset was divided into training (80%) and testing (20%) sets to evaluate model performance. This division ensured that the model could be trained on a substantial portion of the data while retaining a separate subset for validation.

The fourth step was the representation of states for the RL agent. The states observed by the RL agent were indexed for ease of computation: self.state[0] for grind size, self.state[1] for brew ratio, self.state[2] for brew time, and self.state[3] for temperature. State values were constrained using a clip function to ensure they remained within defined upper and lower bounds, which were derived from baristas' brewing data for the medium roasting level.

The fifth step involved encoding actions as discrete values. Actions were encoded as follows: a = 0 (decrease the variable value), a = 1 (maintain the variable value), and a = 2 (increase the variable value). This encoding allowed the agent to make precise adjustments to the brewing parameters. This discrete framework simplifies decision-making while allowing precise control over the brewing parameters. State transitions follow a sequential process. Starting from the grind size state, the agent takes an action to either increase, maintain, or decrease the value. After completing this step, the process transitions to the next variable, brew ratio, followed by brew time, and finally temperature. This ordered approach ensures systematic optimization of all variables.

The sixth and final step was the calculation of rewards. Rewards were calculated based on predefined optimal conditions for each variable. Positive rewards were assigned when actions aligned with optimal conditions, while no reward was given for suboptimal actions. This reward structure guided the agent's learning process, encouraging it to refine its policy iteratively.

## Reinforcement learning framework

RL is employed in this study to optimize the extraction variables for pour-over coffee by analyzing each state in the brewing process. The RL agent determines the actions for each variable based on observations of the current state, as shown in Fig. 1. This approach allows for dynamic adjustments to key variables to enhance the overall extraction quality.

The transition model, which defines how the system evolves, is based on MDP, a framework used in reinforcement learning to describe an environment using states, actions, transition probabilities, and rewards. It enables an agent to make sequential

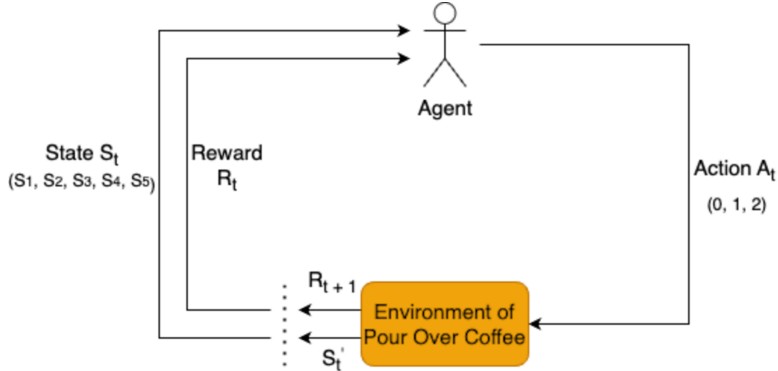

**Figure 1 Reinforcement learning framework for pour-over coffee preparation.**

decisions under uncertainty by maximizing long-term cumulative rewards. This model is expressed as (*Sutton & Barto, 2018*):

$$P(s'|s, a) = \begin{cases} 1, & \text{if } s' = clip(s + (a - 1)) \\ 0, & \text{otherwise} \end{cases}.$$

Here, $P(s'|s, a)$ represents the probability of transitioning from the current state $s$ to the next state $s'$ based on action $a$. Valid transitions, where the next state $s'$ is within the allowable bounds, are rewarded with a probability of 1, while invalid transitions are penalized with a probability of 0. This model provides a robust mechanism for the agent to iteratively refine the brewing parameters.

The transition process is deterministic, where each action leads to a specific next state. For example, an initial state transitions to the next state when the action value is reduced by 1. This is further illustrated in Fig. 2, showing the sequential transitions between states and the systematic refinement of the brewing variables.

Each variable in self.state is updated dynamically based on the action taken by the RL agent. The update process involves adding the value of action−1 to the current state value, reflecting the agent's decision to decrease, maintain, or increase the variable. The specific outcomes are as follows:

(1) If action = 0, then action−1 = −1, causing the state value to decrease by 1.

(2) If action = 1, then action−1 = 0, leaving the state value unchanged.

(3) If action = 2, then action−1 = 1, resulting in the state value increasing by 1.

For instance, if the state value for temperature is 88 °C and the agent selects action = 0, the temperature decreases to 87 °C. Selecting action = 1 keeps the temperature constant at 88 °C, while action = 2 increases the temperature to 89 °C. In the case of brew time, each action modifies the time in increments or decrements of 5 s, depending on the action, ensuring changes are consistent with the brewing process. This method of state adjustment is integral to optimizing the brewing parameters to maximize rewards.

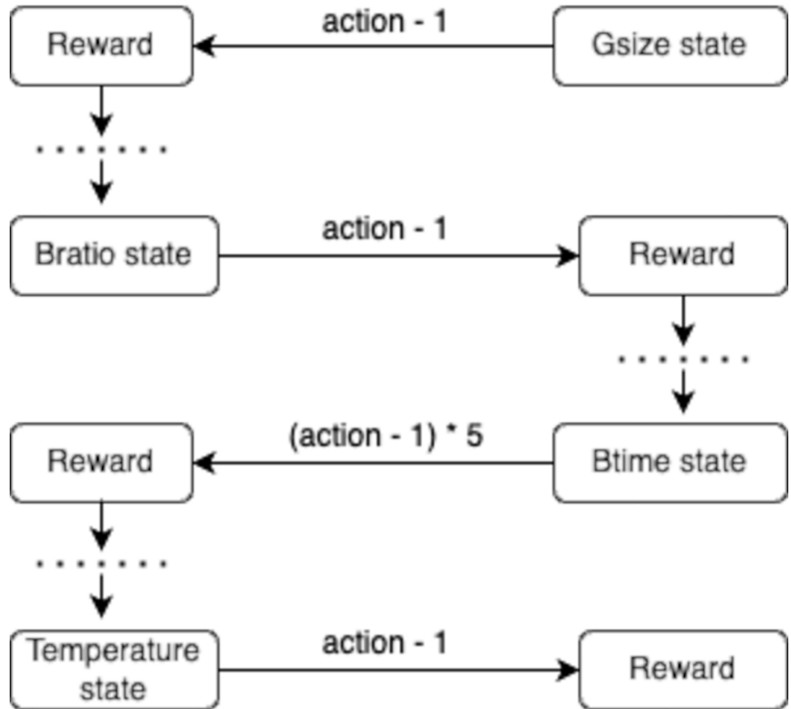

**Figure 2** Flow diagram illustrating the model transition between state variables (*e.g.*, Gsize state, Bratio state, Btime state, Temperature state) and corresponding rewards.

The reward function is pivotal in guiding the agent's learning and decision-making process. After the agent performs an action and transitions to the next state, the associated reward is evaluated and stored in a replay buffer. Positive experiences, defined as those yielding a reward ≥3, are retained to encourage the agent to repeat successful behaviors. Negative experiences, with a reward of 0, are also stored to discourage unproductive actions. This selective storage mechanism enables the agent to refine its policy iteratively, ensuring it learns from both successful and unsuccessful actions.

The replay buffer is designed to hold a finite number of experiences, ensuring the most recent and relevant transitions are prioritized during training. As shown in Fig. 3, the buffer's capacity is set to store 2,000 experiences in this study. When the buffer becomes full, older experiences are discarded following a first in first out (FIFO) approach. This mechanism maintains a dynamic repository of learning experiences, providing the agent with a rich dataset to improve its decision-making capabilities.

The selection of hyperparameters such as buffer size (2,000), learning rate (0.01), and discount factor (0.95) was based on values commonly reported in previous RL studies in similar problem domains. The buffer size of 2,000 was chosen to maintain a manageable memory size while retaining sufficient past experiences to train the agent effectively without overfitting. The learning rate and discount factor were selected to balance convergence speed and learning stability during training.

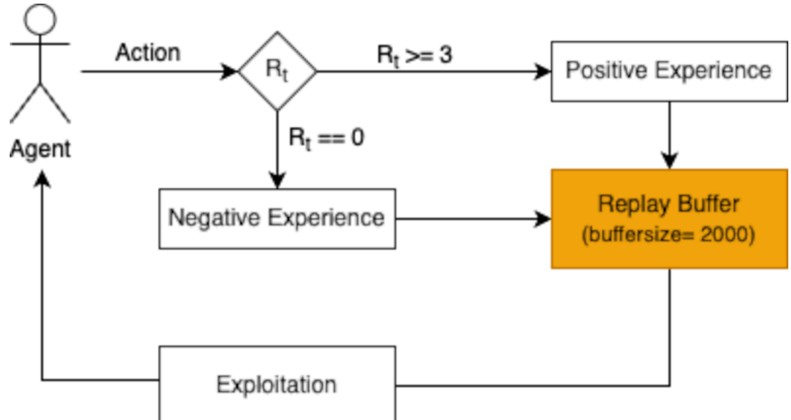

**Figure 3** The replay buffer (with a buffer size of 2,000) stores these experiences for later use in the exploitation phase, allowing the agent to learn effectively from prior interactions.

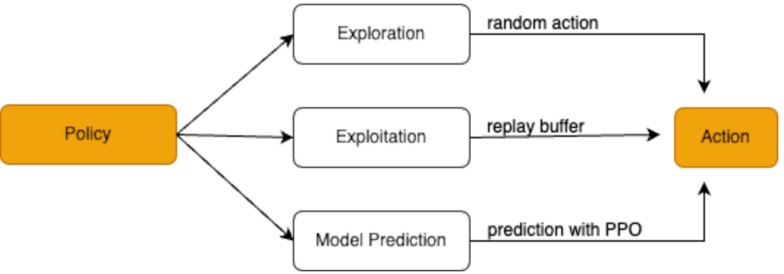

**Figure 4** Policy decision-making process in reinforcement learning.

## RL strategies

The research employs a combination of exploration, exploitation, and model prediction strategies to determine the agent's actions. These strategies, illustrated in Fig. 4, work together to help the agent effectively navigate the trade-offs between trying new approaches and utilizing known successful ones.

The exploration strategy is implemented by allowing the agent to select actions randomly, enabling it to discover various possibilities and maximize rewards. This exploration process is controlled using an epsilon-greedy approach, which adjusts the likelihood of random action selection over time. In this strategy, the agent chooses a random action with a small probability $\varepsilon$ (epsilon) to explore the environment, and chooses the best-known action with a probability $1 - \varepsilon$ to exploit prior learning. This balance between exploration and exploitation is crucial for effective learning. The adjustment follows the formula:

If $\epsilon > \epsilon_{\min}$ then $\epsilon = \epsilon \times \epsilon_{\text{decay}}$

where $\epsilon$ represents the exploration rate, $\epsilon_{min}$ is the minimum allowable exploration rate,

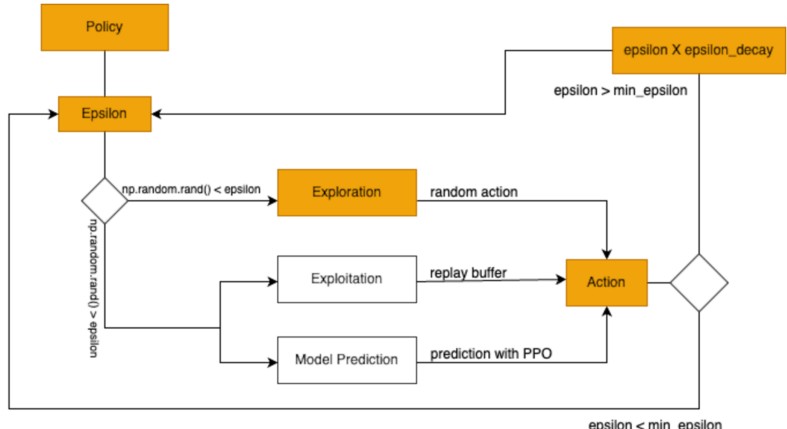

**Figure 5** **The agent engages in exploitation by choosing the action predicted by the model.**

and $\epsilon_{decay}$ is the factor by which $\epsilon$ decreases during each episode. For non-experts, this equation can be thought of as a way to balance experimentation and reliability in brewing. Imagine a barista trying new brewing techniques: at first, they might experiment a lot (high $\epsilon$), like trying different water temperatures or brew times randomly, to see what works best. Over time, as they gain experience, they experiment less (lower $\epsilon$) and rely more on what they have learned.

Initially, $\epsilon$ starts at a higher value to encourage extensive exploration. As episodes progress, $\epsilon$ gradually decreases, reducing the randomness of actions and favoring more informed decisions. When a randomly generated value is less than $\epsilon$, the agent performs an action randomly, ensuring diverse experiences. This process is formalized as:

$$\pi(s) = random\ sample\ a.$$

The implementation of this epsilon-greedy exploration approach in the study is depicted in Fig. 5. As $\epsilon$ decays, the agent transitions from exploration to a more deterministic mode of operation.

If the random value exceeds $\epsilon$, the agent switches to exploitation, utilizing its past experiences stored in the replay buffer. The replay buffer acts as a repository of the agent's historical interactions, storing key variables such as *replay_state* and *replay_action*. During exploitation, the agent retrieves samples from the replay buffer and compares *replay_state* with the current state. If the states align, the agent adopts *replay_action*, which represents the action previously associated with success in a similar state. This process ensures that the agent leverages its accumulated knowledge effectively. Figure 6 illustrates the exploitation mechanism.

The final action selection strategy involves model prediction. A pre-trained model is employed to predict the optimal action for the current state. The model uses the input state to generate a predicted action through the function *model.predict*(*state*). The resulting *predicted_action* guides the agent's decision-making process, providing a data-driven

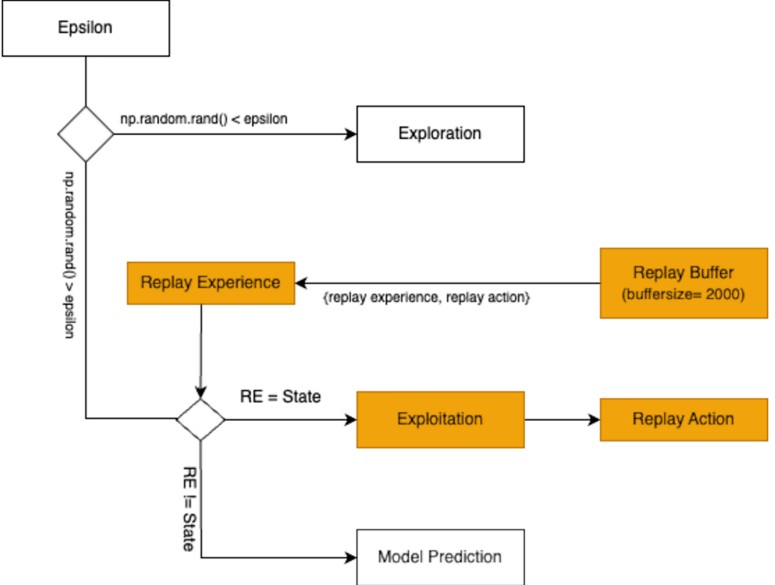

**Figure 6 A flowchart demonstrating the experience replay mechanism in reinforcement learning, where interactions are stored in a replay buffer and sampled for training to improve stability and efficiency during exploitation.**

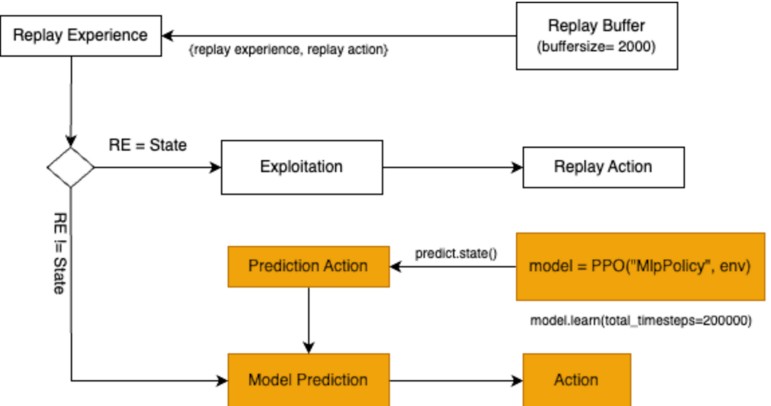

**Figure 7 A flowchart illustrating the prediction, model training, and action-selection process within a reinforcement learning environment.**

alternative to random exploration or reliance on replay buffer samples. The workflow for action selection through model prediction is presented in Fig. 7.

In this study, immediate rewards are assigned to the RL agent based on its ability to take actions that align with predefined optimal conditions. If the action taken matches the expected outcome for the current state, the agent receives a reward of 1. Otherwise, no reward is given. The cumulative reward for each episode is determined by summing the rewards earned across all states or variable values encountered during that episode. This approach ensures that the agent is incentivized to perform optimally at each step of the

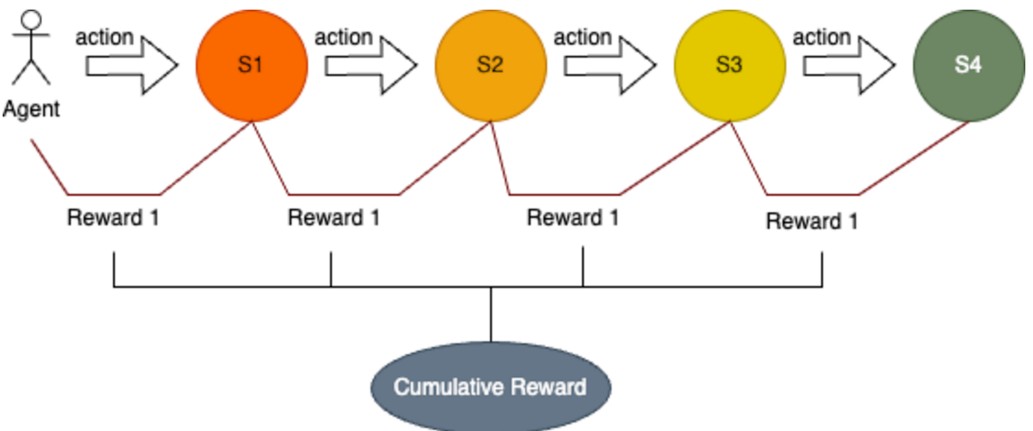

**Figure 8 Sequential decision-making process with state transitions and cumulative reward.**

pour-over coffee brewing process. Figure 8 illustrates the rewards accrued when the agent performs correct actions across all states.

The reward structure is defined through a reward function that evaluates the agent's actions based on the conditions of the states. The rules for calculating rewards are represented by the following formulas:

$$R(s, a, s') = \begin{cases} 1, & \text{if } S_1 = 3 \text{ and } 12 \leq S_2 \leq 14 \text{ and } 120 \leq S_3 \leq 180 \text{ and } 86 \leq S_4 \leq 88 \\ 1, & \text{if } S_1 = 4 \text{ and } 14 \leq S_2 \leq 16 \text{ and } 150 \leq S_3 \leq 180 \text{ and } 88 \leq S_4 \leq 90 \\ 1, & \text{if } S_1 = 5 \text{ and } 16 \leq S_2 \leq 18 \text{ and } 210 \leq S_3 \leq 240 \text{ and } 90 \leq S_4 \leq 92 \\ 0, & \text{otherwise} \end{cases}$$

In the above formula:

- $S_1$ represents the grind size state, with values corresponding to Fine to Medium ($S_1 = 3$), Medium ($S_1 = 4$), and Medium to Coarse ($S_1 = 5$).
- $S_2$ represents the brew ratio, expressed in specific ranges for each grind size.
- $S_3$ is the brew time in seconds, also defined within specific ranges.
- $S_4$ is the temperature, constrained within a narrow optimal range.

The reward function reflects the specific requirements for medium roast coffee and adjusts the conditions for each grind size. For non-experts, this reward function acts like a coffee quality checklist. It checks if the brewing parameters—grind size, brew ratio, brew time, and temperature—fall within ideal ranges for a good cup of coffee, based on barista expertise and SCA standards. For instance, fine to medium grind size ($S_1 = 3$) requires a brew ratio between 12 and 14, a brew time between 120 and 180 s, and a temperature between 86 °C and 88 °C. Similarly, medium grind size ($S_1 = 4$) and medium to coarse grind size ($S_1 = 5$) have their respective ranges for these variables.

The reward system operates sequentially. The agent begins at state $S_1$ and performs an action. If the action meets the reward function criteria, a reward of 1 is granted. Otherwise, the agent receives no reward for that state. The process then transitions to state $S_2$, and the

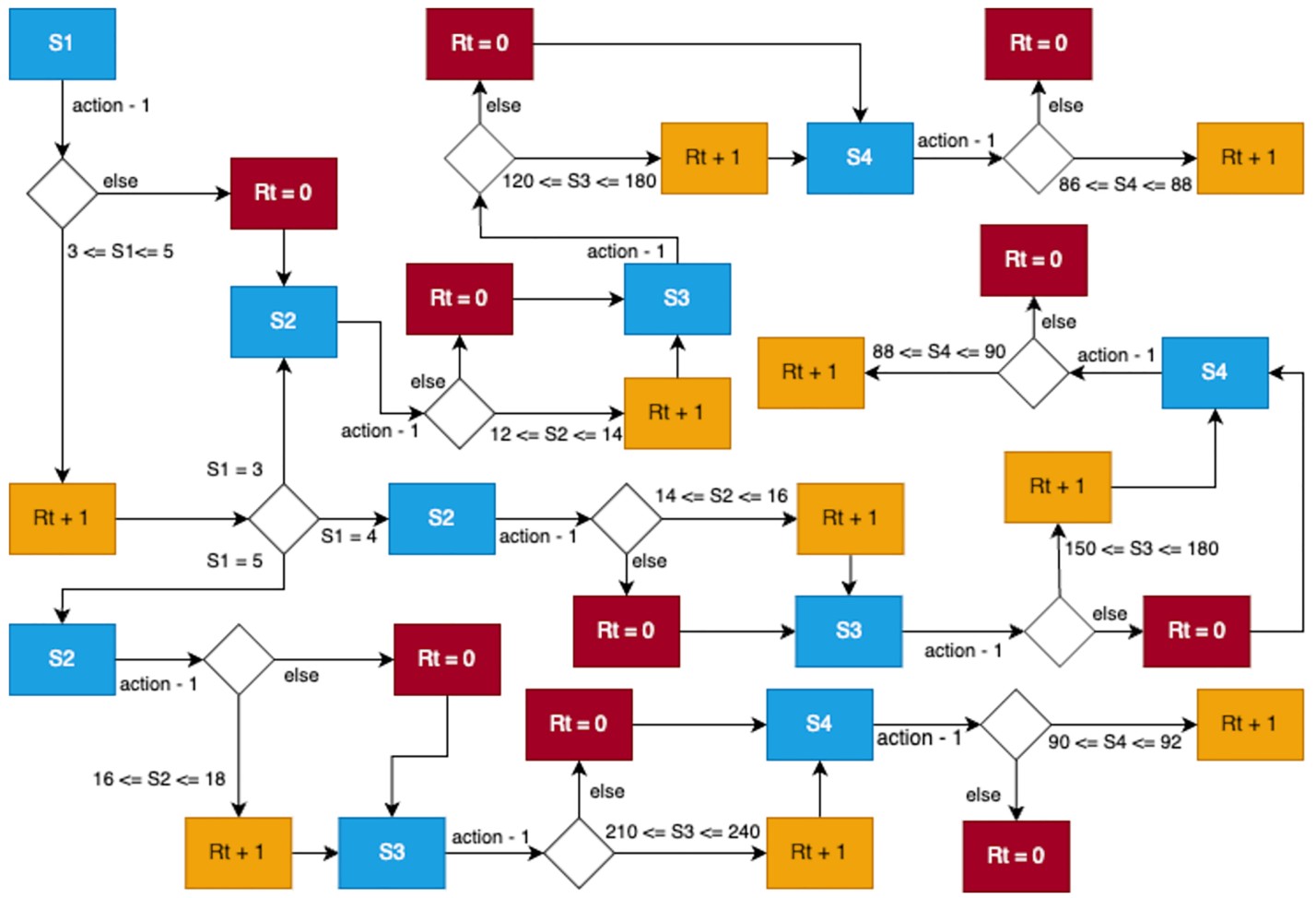

**Figure 9 State transitions with corresponding reward function.**

cycle continues through states $S_4$. If the agent performs correctly for all states, the maximum cumulative reward for an episode is 4.

Figure 9 provides a detailed visualization of the reward implementation, illustrating how the agent navigates through states, takes actions, and accumulates rewards based on its adherence to the predefined reward function. This structured reward system serves as a critical component in guiding the agent toward optimal performance in pour-over coffee extraction.

To address critiques of RL reward design, particularly the challenge of sparse rewards in dynamic systems, we note that the binary reward structure (1 or 0) may lead to sparse feedback, as the agent only receives a reward when all conditions are met. Sparse rewards can hinder learning in dynamic systems like coffee brewing, where variables such as brew time and temperature interact dynamically, and optimal states may be infrequently reached during early training, as highlighted in recent critiques (*Elsayed et al., 2024*; *Yan, Luo & Xu, 2024*). *Yan, Luo & Xu (2024)* argue that sparse rewards in multi-goal navigation tasks (a type of dynamic system) lead to inefficient exploration due to long-sequence

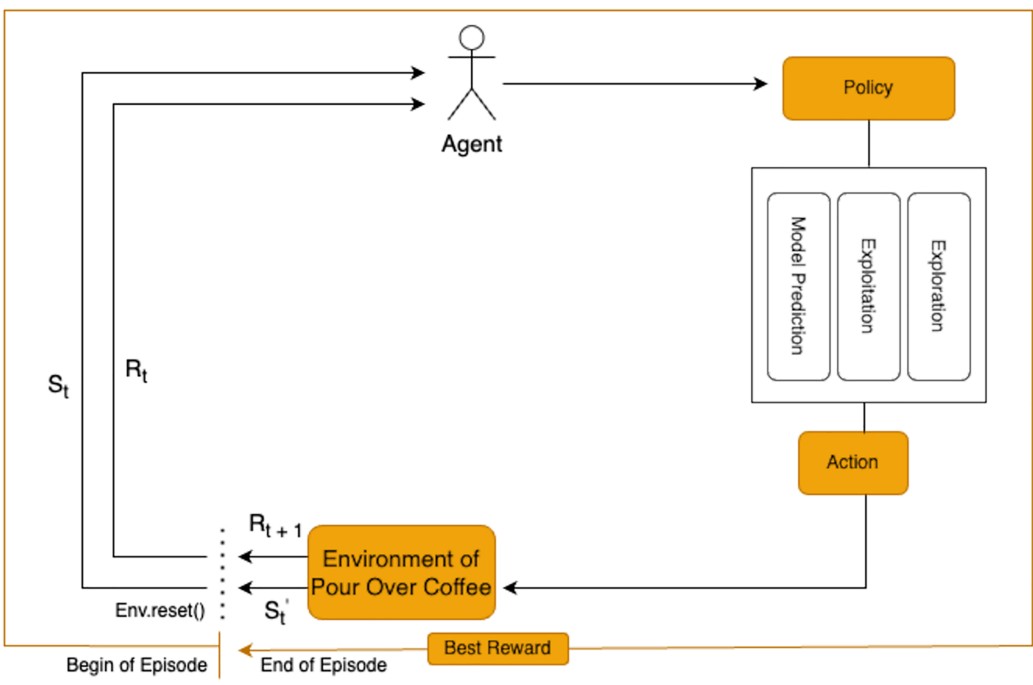

**Figure 10 Reinforcement learning training for optimizing actions in a pour-over coffee environment.**

decision-making, while (*Elsayed et al., 2024*) note that sparse rewards in robotic control tasks can cause unstable learning. While this approach facilitated effective learning in our experiments (as evidenced by the RL model's high performance), future work could explore reward shaping—*e.g.*, providing intermediate rewards for partial alignment with optimal conditions—or intrinsic rewards like curiosity-driven exploration to further address the sparse reward challenge in dynamic brewing systems, aligning with best practices in RL research.

## RL training

The RL training process spanned 60 episodes, where each episode began with resetting the environment and observing the initial state. The agent then selected actions based on its policy. Correct actions yielded rewards and transitioned the agent to the next state, which reflected the effect of the action taken. At the end of each episode, the agent determined the best reward achieved during that episode, as visualized in Fig. 10.

Although Stable-Baselines3 was reviewed and evaluated during early experimentation, its default agents required additional modification to support our custom environment and multi-variable state encoding structure. As a result, we opted for a lightweight custom implementation of RL tailored to the pour-over coffee environment. This allowed more precise control over state-action definitions, reward shaping, and episode resets. We acknowledge that future extensions of this work could benefit from benchmarking against standardized agents available in frameworks such as Stable-Baselines3 for broader reproducibility and performance comparison.
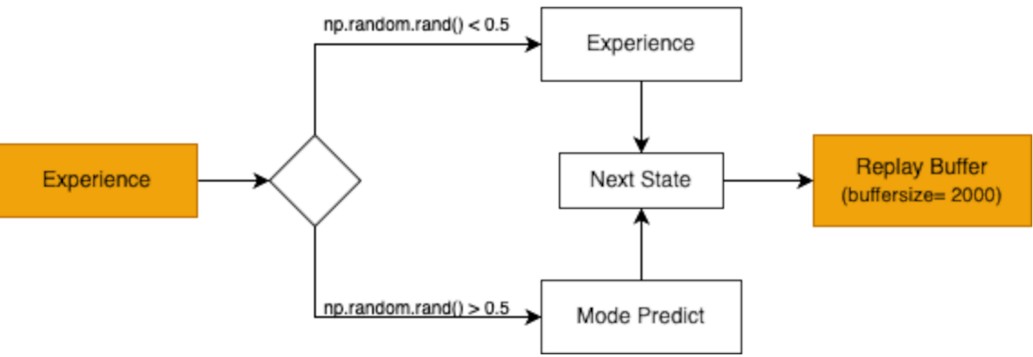

**Figure 11 Experience replay mechanism with a buffer size of 2,000 for storing experiences and predicting the next state.**

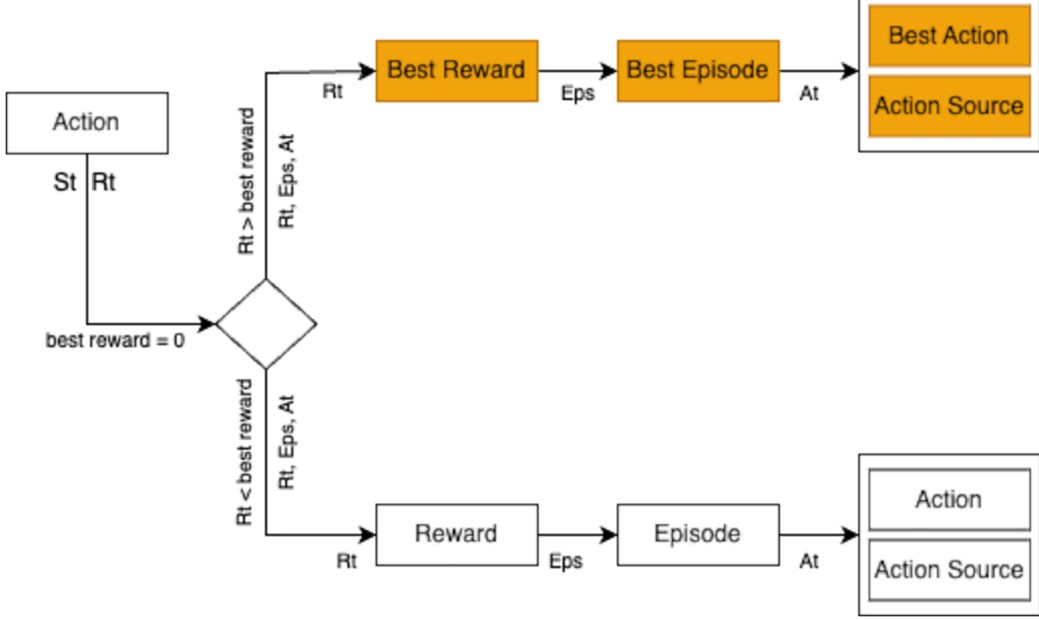

**Figure 12 Selection process for the best action and reward evaluation within episodes.**

Prior to training, a prediction model was built to guide the agent's actions effectively. This preparatory phase enabled the model to predict actions and gauge its understanding of the environment. Subsequently, the replay buffer was populated with experiences from exploration and model prediction phases, as shown in Fig. 11. These experiences enriched the agent's knowledge base, setting the stage for effective decision-making during RL.

During training, the agent employed three action selection strategies: exploration, exploitation, and model prediction. The strategy yielding the highest reward in an episode was chosen. For instance, if exploration yielded a reward of 2, exploitation a reward of 3, and model prediction a reward of 4, the model prediction action was selected due to its superior reward, as depicted in Fig. 12.

**Table 4 Sample results of reinforcement learning training.**

| Episode | Score | Best reward | Gsize state | Bratio state | Btime state (min:sec) | Temperature state | Best action source |
|---|---|---|---|---|---|---|---|
| 46 | 160 | 4 | 5 | 1:16 | 4 m | 92 | Model prediction |
| 47 | 127 | 4 | 4 | 1:14 | 2 m 55 s | 90 | Model prediction |
| 48 | 143 | 4 | 3 | 1:14 | 2 m 10 s | 88 | Exploration |
| 49 | 170 | 4 | 3 | 1:14 | 2 m | 88 | Model prediction |
| 50 | 145 | 4 | 5 | 1:17 | 4 m | 90 | Exploitation |
| 51 | 169 | 4 | 3 | 1:14 | 2 m 10 s | 86 | Exploration |
| 52 | 146 | 4 | 5 | 1:16 | 4 m | 92 | Exploitation |
| 53 | 156 | 4 | 5 | 1:16 | 4 m | 90 | Exploration |
| 54 | 139 | 4 | 3 | 1:13 | 2 m 10 s | 88 | Exploration |
| 55 | 111 | 3 | 4 | 1:14 | 2 m | 90 | Model prediction |
| 56 | 194 | 4 | 4 | 1:14 | 2 m 50 s | 88 | Exploration |
| 57 | 181 | 4 | 4 | 1:15 | 2 m 45 s | 90 | Exploration |
| 58 | 172 | 4 | 3 | 1:13 | 2 m 15 s | 87 | Exploration |
| 59 | 194 | 4 | 4 | 1:15 | 2 m 45 s | 89 | Model prediction |
| 60 | 136 | 4 | 3 | 1:13 | 2 m 5 s | 88 | Exploration |

The training results are summarized in Table 4, showcasing data from 15 episodes out of the total 60. Episodes with a reward of 4 indicate that all actions conformed to the reward function, while episodes with a reward of 3 suggest that one action failed to meet the criteria. The cumulative score for each episode reflects the total rewards obtained, with the action source indicating whether it stemmed from exploration, exploitation, or model prediction. A stable learning curve for the RL model is observed in the graph of total scores per episode, shown in Fig. 13.

For KNN data generation, 3,600 data points were simulated using the pour-over coffee environment. These data were divided into training (80%) and testing (20%) sets. The initial states were generated *via* the *env*() function, and actions were determined based on KNN model predictions, as illustrated in Fig. 14. The KNN model selected the three nearest neighbors for training, spanning the same 60 episodes as RL. Each episode began with resetting the environment and initializing the state into a tuple, following the implementation outlined in Fig. 15. In contrast, the KNN algorithm resembles a barista consulting a notebook of past brews. When faced with a new situation, it finds similar past brewing scenarios and chooses the action that previously yielded the best result. Unlike reinforcement learning, KNN does not learn from experience—it simply recalls and imitates successful outcomes.

All features were normalized to ensure equal scaling, preventing any bias in distance calculations. The Euclidean distance metric was used to identify the nearest neighbors. Unlike RL, the KNN model served as a static, memory-based baseline that did not update its policy during training, enabling a direct performance comparison with the adaptive RL agent. During KNN training, the actions relied solely on predictions from the model. Similar to RL, the best reward from each iteration was identified as the episode's optimal

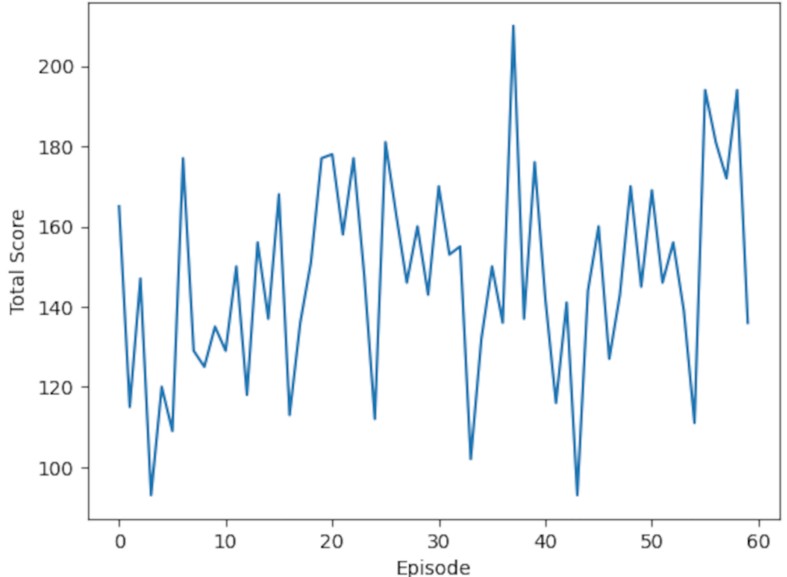

**Figure 13 Learning curve for RL model: the plot displays the model's performance (Total score, range 0–60) across training episodes (0–60).** Scores are annotated at intervals of 10. The curve reflects the model's learning progression over these episodes.

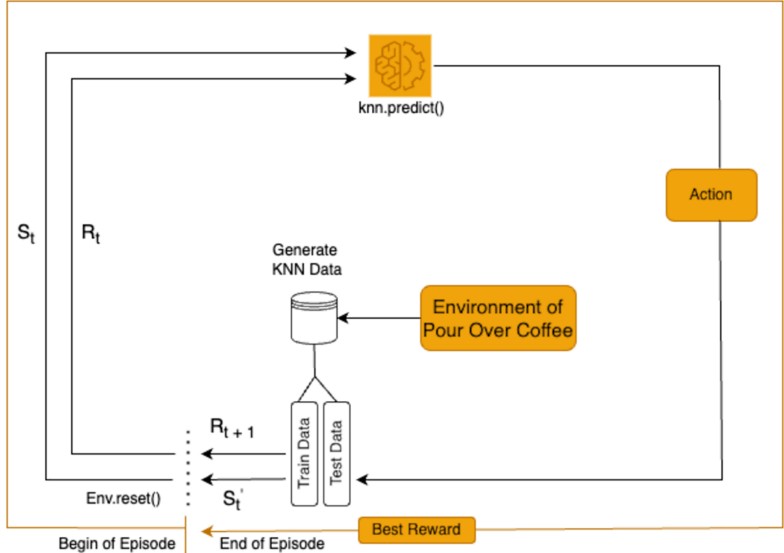

**Figure 14 KNN is integrated into the system for decision-making or state-action prediction.** The system iteratively interacts with the environment, utilizing KNN to refine performance toward optimal results.

reward. However, unlike RL, KNN exclusively used model predictions as the action source. The process for selecting the best reward is depicted in Fig. 16.

The training results yielded insights into rewards, scores, and states. Table 5 presents a sample of KNN training outcomes. Episodes with a reward of 3 indicated that one action

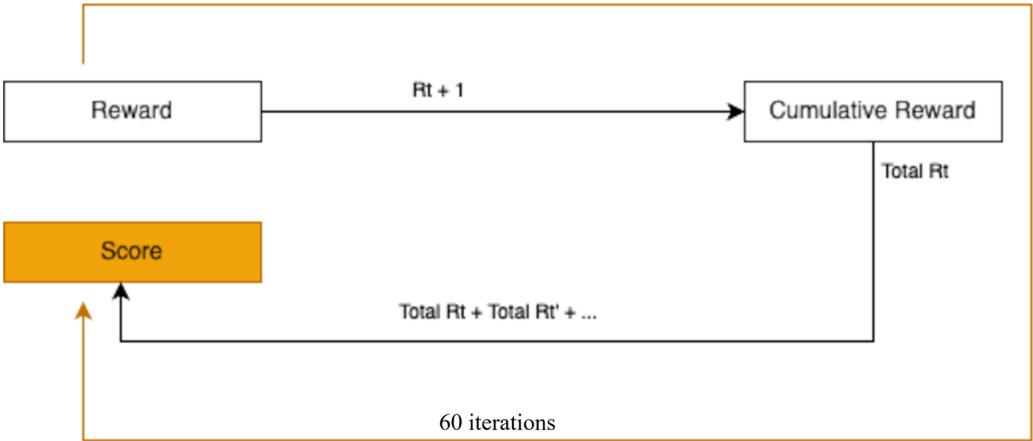

**Figure 15** The KNN algorithm undergoes training over multiple episodes to enhance its performance based on cumulative learning outcomes.

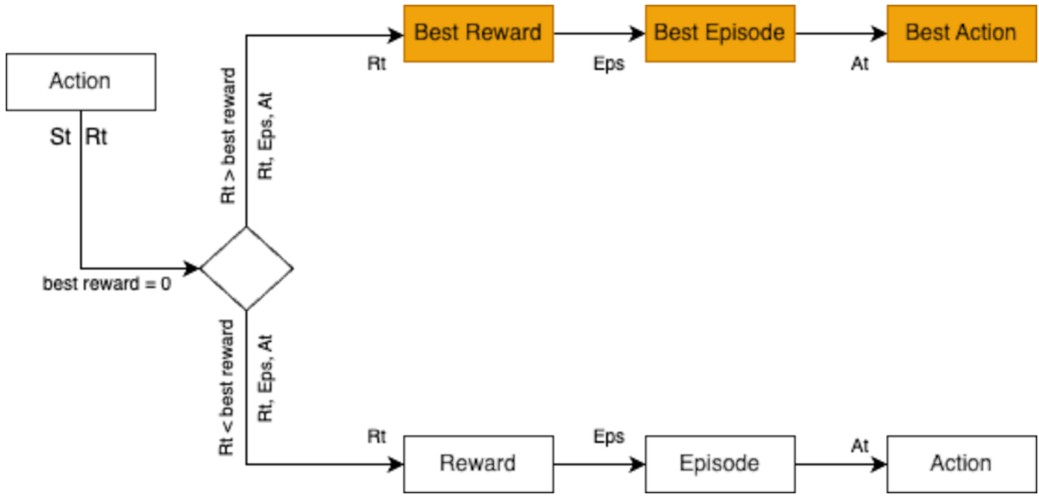

**Figure 16** The KNN model identifies the best reward across multiple episodes by evaluating actions and rewards to optimize performance.

deviated from the optimal state, while episodes with a reward of four demonstrated accurate predictions for all states. These results were further analyzed through a graph illustrating the total scores per episode, which provides a visual representation of the KNN model's learning curve in optimizing pour-over coffee extraction variables. This learning curve is shown in Fig. 17.

## RESULTS

The results of the study highlight the performance of RL and KNN in optimizing pour-over coffee extraction variables. Table 6 presents the rewards achieved over 60 episodes, showing RL secured eight rewards of 3 and 52 rewards of 4, while KNN attained

| Table 5 Sample results of KNN training. | | | | | | |
|---|---|---|---|---|---|---|
| Episode | Score | Reward | Gsize state | Bratio state | Btime state (min:sec) | Temperature state |
| 46 | 121 | 4 | 3 | 1:13 | 2 m 15 s | 87 |
| 47 | 149 | 4 | 5 | 1:16 | 4 m | 91 |
| 48 | 135 | 4 | 4 | 1:15 | 2 m 40 s | 89 |
| 49 | 114 | 3 | 5 | 1:16 | 2 m 10 s | 90 |
| 50 | 180 | 4 | 4 | 1:15 | 2 m 30 s | 90 |
| 51 | 140 | 4 | 5 | 1:16 | 4 m | 90 |
| 52 | 147 | 4 | 5 | 1:16 | 4 m | 91 |
| 53 | 112 | 4 | 3 | 1:14 | 2 m | 88 |
| 54 | 144 | 4 | 4 | 1:16 | 2 m 55 s | 90 |
| 55 | 130 | 4 | 4 | 1:15 | 2 m 30 s | 90 |
| 56 | 132 | 4 | 3 | 1:12 | 2 m 5 s | 88 |
| 57 | 118 | 4 | 5 | 1:16 | 3 m 55 s | 92 |
| 58 | 110 | 4 | 3 | 1:14 | 2 m 40 s | 86 |
| 59 | 109 | 4 | 3 | 1:12 | 2 m | 86 |
| 60 | 116 | 4 | 5 | 1:16 | 3 m 35 s | 91 |

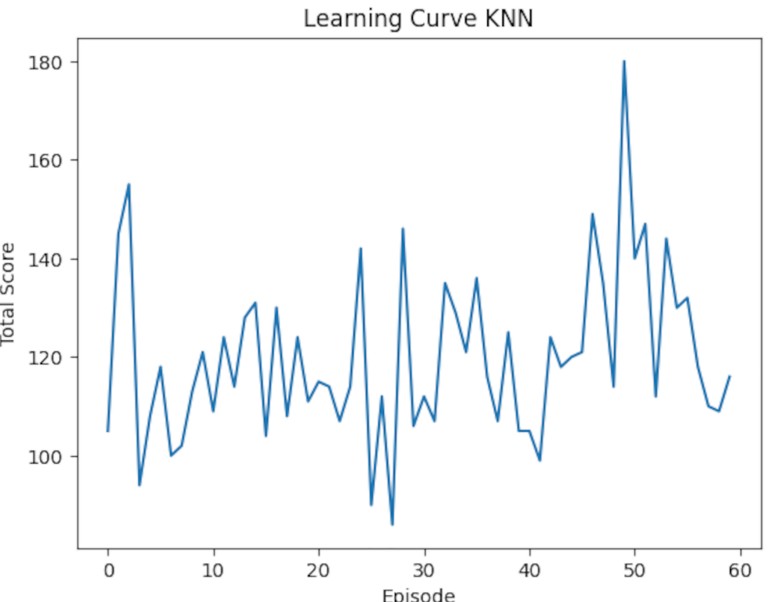

**Figure 17 The KNN learning curve shows the variation in total score across episodes, highlighting performance trends and fluctuations throughout the training process.**

12 rewards of 3 and 48 rewards of 4. This indicates that RL slightly outperformed KNN in achieving the highest reward frequency.

Table 7 further compares the average rewards and scores of both models. RL demonstrated a marginally higher average reward of 3.87 (95% CI [3.74–4.00]) compared to KNN's 3.80 (95% CI [3.67–3.93]). However, the difference in the average scores was

**Table 6 Distribution of rewards for RL and KNN, showing that both models achieve no rewards for levels 1 to 3, while RL secures higher rewards at level 4 (8 *vs.* 12) and level 5 (52 *vs.* 48) compared to KNN.**

| Reward | RL | KNN |
|---|---|---|
| 1 | 0 | 0 |
| 2 | 0 | 0 |
| 3 | 0 | 0 |
| 4 | 8 | 12 |
| 5 | 52 | 48 |

**Table 7 Average reward and score comparison between RL and KNN, showing RL achieves a higher average reward (3.87) and score (146.93).**

| Average | RL | KNN |
|---|---|---|
| Reward | 3.87 | 3.80 |
| Score | 146.93 | 119.87 |

more pronounced, with RL achieving 146.93 (95% CI [141.87–151.99]), surpassing KNN's 119.87 (95% CI [114.57–125.17]). To assess statistical significance, paired t-tests were conducted on the per-episode rewards and scores. The difference in average rewards was statistically significant ($p = 0.038$), as was the difference in average scores ($p < 0.001$). Confidence intervals (95%) were computed using the standard error of the mean over the 60 episodes, assuming a normal distribution, to provide a range within which the true metric values are likely to lie. The reward comparison is illustrated in Fig. 18, which underscores RL's consistent advantage in several episodes, though KNN outperformed it occasionally. Meanwhile, Fig. 19 shows the score comparison, highlighting that KNN generally lagged behind RL across most episodes.

## Testing brew results

The optimized extraction variables derived from RL were tested using pour-over coffee equipment and ingredients. The quality of the brewed coffee was evaluated by measuring the percentage of TDS using a coffee refractometer. The optimal brewing parameters, based on episode 46, included:

- Grind size: 5 (medium-coarse)
- Brew ratio: 1:16
- Brew time: 4 min
- Temperature: 92 °C

Using 18 grams of coffee, the total water volume was calculated as $18 \times 16 = 288$ ml. Post-brewing, the total brewed coffee volume was 258 ml, accounting for evaporation and the retained mass in the coffee grounds. The refractometer measured a TDS percentage of 1.29%, which falls within the SCA optimal range of 1.15–1.55%.

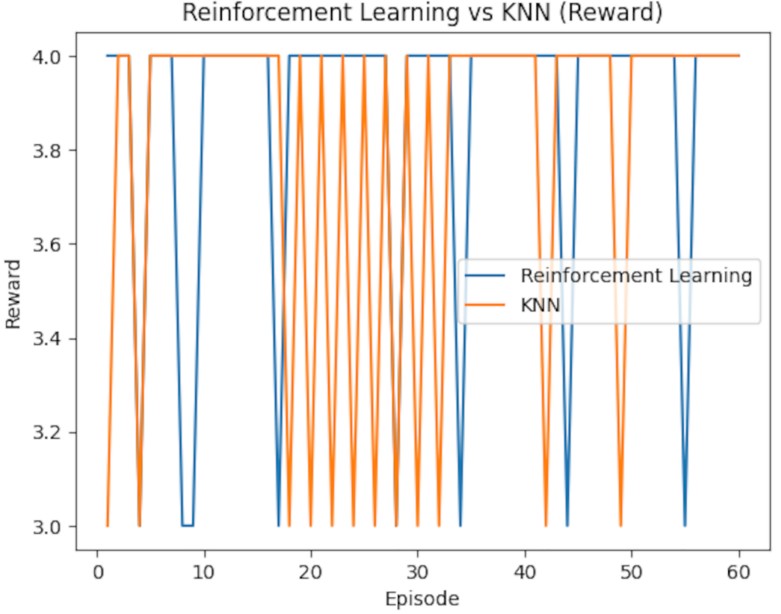

**Figure 18** **Learning curve for KNN model: the plot displays the model's performance (Total score, range 0–60) across training episodes (0–60).** Scores are annotated at intervals of 10. The curve reflects the model's learning progression over these episodes.

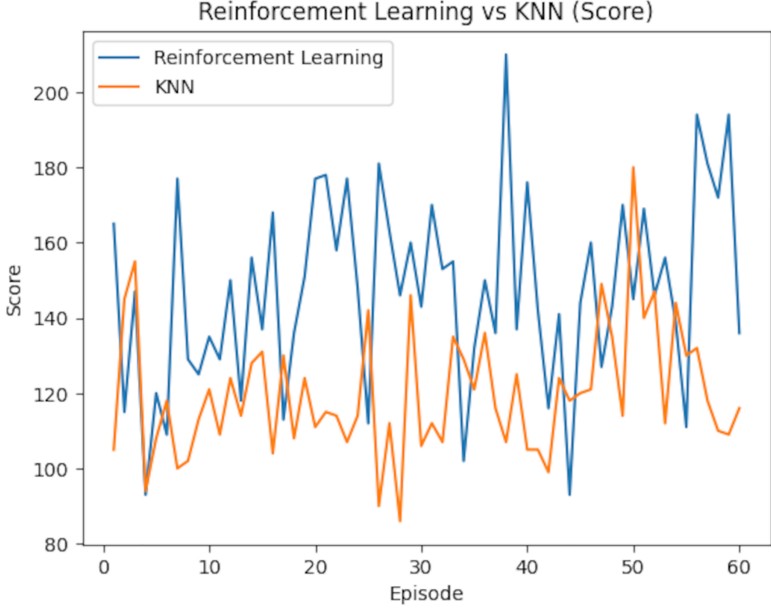

**Figure 19 Performance comparison between reinforcement learning and KNN models over episodes, showing variations in score during training.**

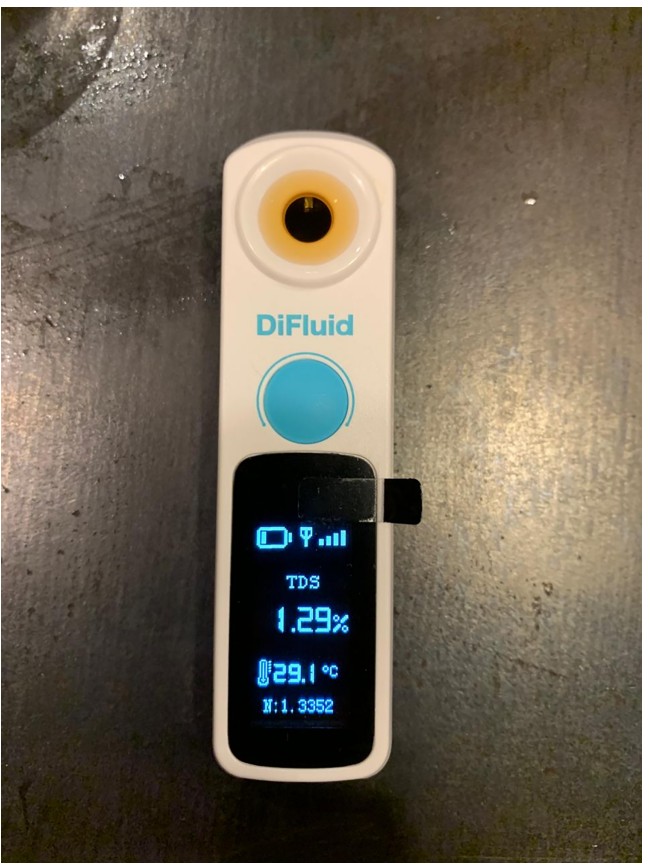

**Figure 20** DiFluid device displaying total dissolved solids (TDS) value of 1.29% at a temperature of 29.1 °C, providing coffee quality measurements.

The extraction yield percentage (EY%) was calculated using the formula (*Lingle, 2011*):

$$\text{EY\%} = \left( \frac{1.29\% \times 258}{18} \right).$$

This value confirms that the extraction fell within the SCA's optimal range of 18–22%. Figure 20 displays the TDS measurement, and Fig. 21 visualizes the extraction yield. The optimal extraction point, indicated within the central box, verifies that the brew from RL's optimized parameters achieved a desirable quality. This result underscores the effectiveness of RL in identifying and optimizing critical pour-over coffee variables for a superior brewing experience.

## Sensory evaluation

To validate the sensory quality of the RL-optimized brew, we conducted a blind taste test and expert evaluation with five professional baristas from Ujala Café & Roastery. In the blind taste test, baristas evaluated two samples: the RL-optimized brew (grind size: 5, brew ratio: 1:16, brew time: 4 min, temperature: 92 °C) and a baseline brew prepared using manually determined parameters by a barista (grind size: 4, brew ratio: 1:15, brew time: 3 min, temperature: 90 °C). The samples were labeled A and B, with the order randomized

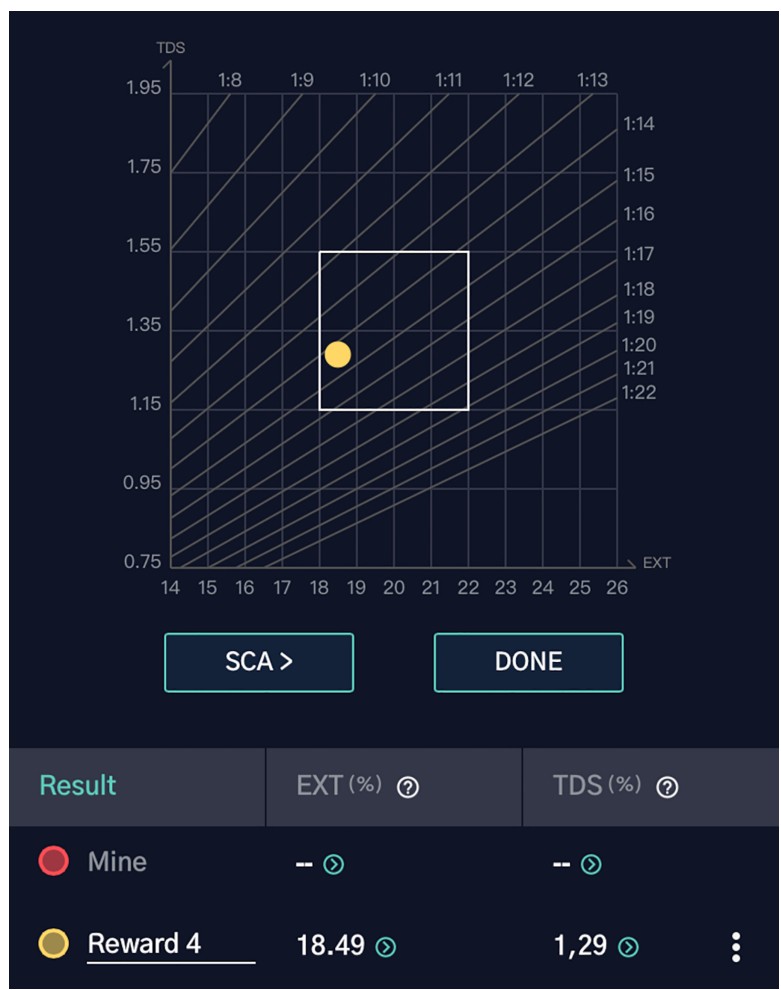

**Figure 21 Results graph showing the measured extraction yield (EXT = 18.49%) and total dissolved solids (TDS = 1.29%) for "Reward 4".**

to eliminate bias. Each barista scored the samples on a 1–10 scale for overall quality, considering attributes like aroma, flavor, acidity, body, and aftertaste. The RL-optimized brew received a mean score of 8.4 (SD 0.8), while the baseline brew scored 7.6 (SD 0.9). A paired t-test indicated that the RL-optimized brew was rated significantly higher ($p = 0.031$).

For expert validation, the same baristas evaluated the RL-optimized brew using a simplified SCA cupping protocol, scoring five attributes—aroma, flavor, acidity, body, and aftertaste—each on a 1–10 scale. The average scores were: aroma 8.6 (SD 0.5), flavor 8.8 (SD 0.4), acidity 8.4 (SD 0.5), body 8.2 (SD 0.6), and aftertaste 8.4 (SD 0.5). The total score, summed across attributes, was 42.4 out of 50 (SD 1.8), equivalent to approximately 84.8/100 on the SCA cupping scale, indicating specialty quality (typically above 80/100). These results confirm that the RL-optimized parameters not only meet objective SCA standards (TDS and extraction yield) but also produce a high-quality brew according to expert sensory evaluation.

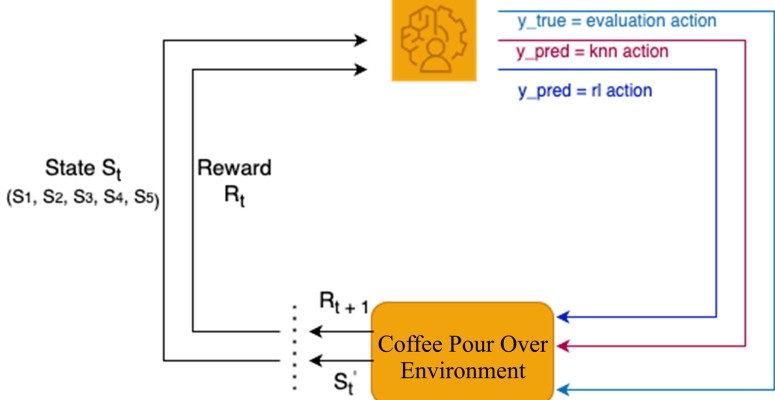

**Figure 22 The predicted and correct actions derived from evaluation function.**

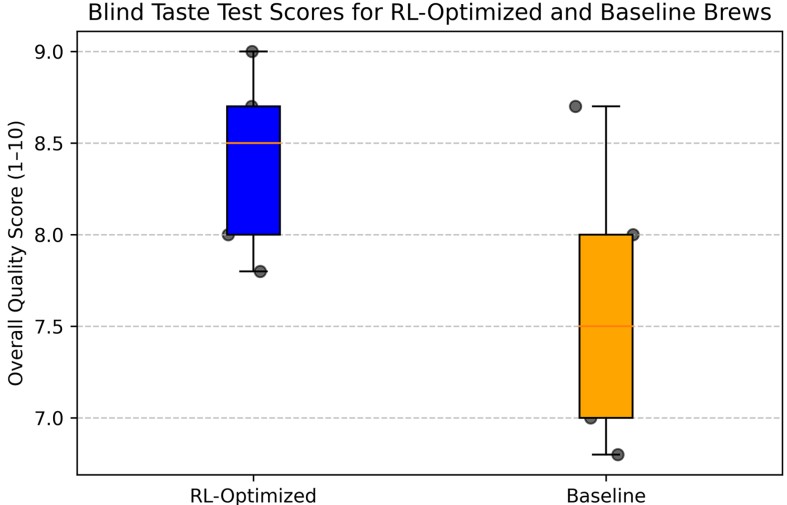

**Figure 23 Box plot comparison of blind taste test scores for RL-optimized and baseline coffee brews.**

Figure 22 illustrates the blind taste test scores, showing the distribution of ratings for the RL-optimized and baseline brews. Figure 23 presents the average scores for each sensory attribute from the expert evaluation, highlighting the balanced profile of the RL-optimized brew across all dimensions. These findings validate the practical effectiveness of RL in optimizing pour-over coffee brewing for sensory quality.

## Evaluation

The evaluation of the RL model and its comparison with the KNN model involved the implementation of an evaluation function. This function was designed to align with the reward function, ensuring that the models' actions were assessed based on their ability to achieve optimal rewards. For instance, if the grind size state equaled 3, the brew ratio state was within the range of 12 to 14, the brew time state was between 120 and 180, and the

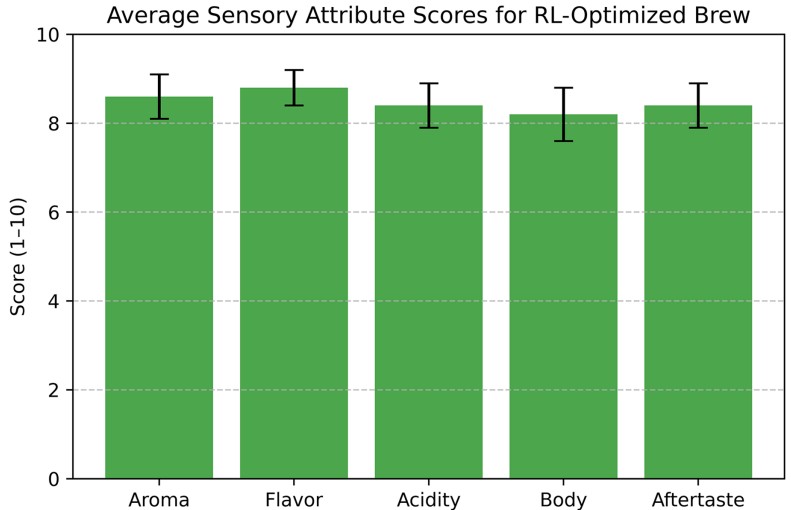

**Figure 24 Average sensory attribute scores for RL-optimized coffee brews with standard deviation error bars.**

**Table 8 Comparison of performance metrics between RL and KNN classifiers, including accuracy, precision, recall, and F1-score.**

| Matrices | RL | KNN |
|---|---|---|
| Accuracy | 90.00% | 88.33% |
| Precision | 90.76% | 90.19% |
| Recall | 90.00% | 88.33% |
| F1-score | 90.08% | 88.90% |

temperature state was between 86 and 88, the function would return an action value of 1, maintaining the current state values. This approach was integrated into the training processes of both models to provide a standardized evaluation mechanism.

As illustrated in Fig. 24, the evaluation process compared $y_{pred}$, representing the predicted actions from the models, with $y_{true}$, which corresponded to the correct actions derived from the evaluation function. The comparison produced performance matrices, which provided insights into the accuracy, precision, recall, and F1-score of each model in optimizing the pour-over coffee extraction variables as shown in Table 8.

The performance analysis revealed that the RL model achieved an accuracy of 90.00% (95% CI [88.73–91.27%]), a precision of 90.76% (95% CI [89.49–92.03%]), a recall of 90.00% (95% CI [88.73–91.27%]), and an F1-score of 90.08% (95% CI [88.81–91.35%]) over the 60 episodes. In comparison, the KNN model demonstrated slightly lower performance, with an accuracy of 88.33% (95% CI [87.00–89.66%]), a precision of 90.19% (95% CI [88.86–91.52%]), a recall of 88.33% (95% CI [87.00–89.66%]), and an F1-score of 88.90% (95% CI [87.57–90.23%]). A paired t-test on the per-episode accuracy values showed a statistically significant difference between RL and KNN ($p = 0.042$). These findings indicate that the RL model outperformed the KNN model in all key evaluation metrics. Although both models displayed high levels of precision and recall, the RL model

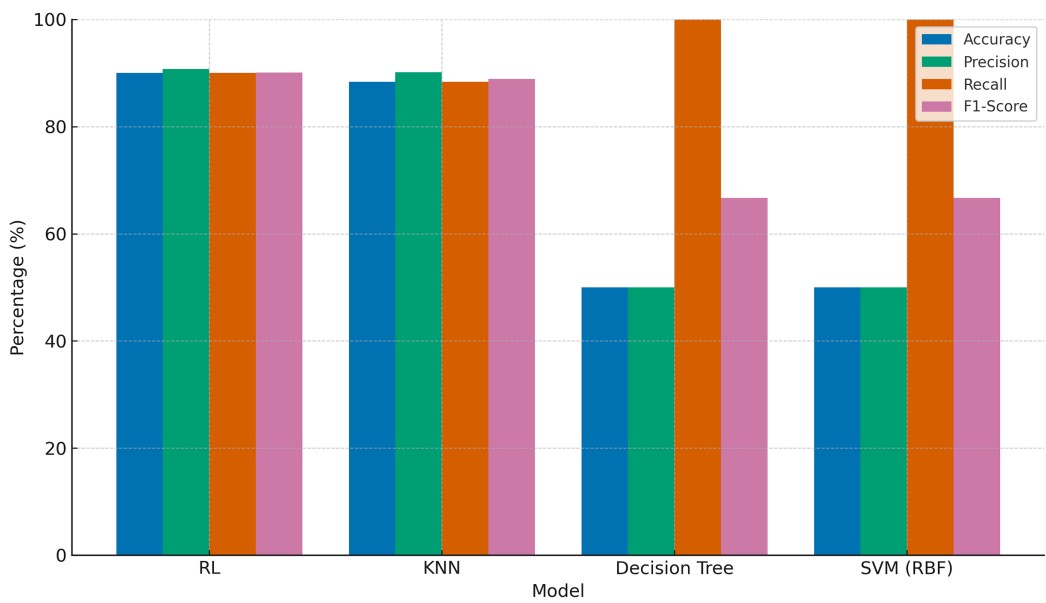

**Figure 25 Comparative performance of RL, KNN, decision tree, and SVM models in optimizing pour-over coffee extraction variables.**

proved to be more consistent and accurate, as reflected in its higher F1-score. This demonstrates the superior capability of RL in effectively optimizing the variables for pour-over coffee extraction.

To provide a broader benchmark, additional models including support vector machine (SVM) and Decision Tree classifiers were evaluated using the same reward-based label schema derived from RL criteria. These models were trained on the same dataset and preprocessing pipeline to ensure consistency. As shown in Fig. 25, RL achieved the highest evaluation scores across all metrics. KNN performed competitively but with slightly lower recall and F1-score. In contrast, SVM and decision tree models, though achieving perfect recall (100%), misclassified all suboptimal brews as optimal, resulting in only 50.00% accuracy and 66.67% F1-score. This discrepancy underscores the limitations of traditional classifiers in dynamic, multivariate optimization scenarios and emphasizes the robustness of RL in such settings.

A simple hyperparameter sensitivity analysis was conducted to examine the effects of varying buffer size (1,000, 2,000, 3,000) and learning rate (0.001, 0.01, 0.1). The model performed best with a buffer size of 2,000 and a learning rate of 0.01, showing improved stability and convergence. Larger buffers slightly increased training time without improving results, while excessively high learning rates led to unstable learning behavior. These findings support the chosen values for this experimental setting.

## DISCUSSION

The findings of this study are consistent with prior research in RL and coffee-related studies. *Huang et al. (2022)*, *Dharmawan & Bintang (2020)*, *Chen, Chiu & Liu, 2021*, *Cahyo & Hayati (2022)*, *Bokade, Jin & Amato (2023)*, *Singh et al. (2022)*, *Ibarz et al. (2021)*,

*Amin et al. (2023)*, *Yau et al. (2023)*, *Fujita, Sato & Nobuhara (2021)*, *Zhang et al. (2022)* highlight RL's applicability in dynamic and complex environments. In the studies by *Anita & Albarda (2020)*, *Hakim, Djatna & Yuliasih (2020)*, *Okamura et al. (2021)*, *Przybył et al. (2023)*, *Yu et al. (2021)*, and *Alamri et al. (2023)* similarities can be observed in the research focus, which is centered on coffee. However, this study distinguishes itself by being the first to apply RL to optimize variables in the pour-over coffee brewing process. Unlike earlier studies that primarily focused on classification tasks such as coffee roasting levels and bean quality or used supervised and unsupervised methods, this research emphasizes dynamic optimization of grind size, brew ratio, brew time, and temperature.

Recent advancements in dynamic optimization further support the potential of reinforcement learning in applications like coffee brewing, where system parameters vary in real time. For example, *Ahmed (2024)* introduced an adaptive metaheuristic framework capable of integrating real-time feedback, enhancing solution stability under dynamic conditions. *Aoyama, Lehmamnn & Theodorou (2024)* developed a second-order Stein variational optimization method that improves trajectory adaptation and mitigates local minima. *Hou (2024)* proposed a zero-shot Lagrangian update technique, enabling online systems to respond rapidly to abrupt changes. *Lei et al. (2024)* advanced a prediction strategy using second-order derivatives for dynamic multi-objective problems, improving the accuracy of future state estimations. These innovations highlight how RL and modern optimization strategies can work in tandem to address challenges in domains that require continuous real-time tuning, such as beverage extraction systems.

RL is particularly suited for dynamic and complex environments but has yet to be applied to pour-over coffee brewing methods. This study introduces RL as a novel approach to determine the complexity of variables in pour-over coffee. Previous research on coffee has primarily centered on classification tasks, such as analyzing roasting levels and bean quality. While certain studies, such as *Yu et al. (2021)*, have explored aspects of pour-over coffee brewing, their emphasis was on utilizing virtual reality for educational purposes. Additionally, most prior research relied on supervised or unsupervised learning techniques, setting them apart from the RL-based methodology proposed in this study.

A significant finding of this research is that RL outperforms KNN in optimizing pour-over coffee extraction variables. This superiority stems from the dynamic nature of the pour-over coffee environment, where the initial state resets at the beginning of each episode. RL provides a broader range of actions through exploration, unlike KNN, which depends on labeled data. By incorporating three action selection strategies, this study optimizes the rewards obtained by the agent.

These results further validate RL's capacity to model dynamic interactions in pour-over coffee brewing, outperforming both static memory-based models like KNN and conventional classifiers like decision trees and SVM. While decision tree and SVM models excelled in identifying optimal brews (high recall), they lacked precision, often failing to differentiate between good and suboptimal extractions. This suggests that static models are less suited for fine-tuned, sequential adjustments required in artisanal brewing tasks. In contrast, the RL agent learned to iteratively adjust multiple interdependent variables, maintaining a more balanced performance profile.

While the results are presented in terms of accuracy and extraction yield, they also offer qualitative insights into practical brewing. For instance, the RL agent's preference for certain combinations—such as higher brew ratios paired with moderate temperatures—suggests a tendency toward maximizing flavor clarity while avoiding over-extraction. This balance may translate into a cup profile that is cleaner and more consistent, particularly useful for commercial or home brewers seeking repeatable quality. Additionally, the RL model's avoidance of extreme grind sizes indicates a learned sensitivity to flow rate and clogging potential, reinforcing the value of human-intuitive strategies learned autonomously.

In terms of real-world applicability, the proposed RL framework offers promising potential for integration into automated or semi-automated coffee brewing systems. The agent's ability to learn optimal brewing sequences based on feedback could be deployed in smart coffee machines, enabling dynamic, user-tailored adjustments in real time. Additionally, the modular architecture of the environment allows for scalability and better latency, making it possible to extend the system to other coffee preparation methods, such as espresso or French press, by simply redefining state variables and reward structures. This adaptability supports broader use in both consumer-grade appliances and industrial coffee settings.

RL's potential scalability for commercial coffee systems lies in its ability to adapt to diverse conditions and user preferences. Pre-trained RL models could be embedded into smart coffee machines, where brewing variables are controlled *via* microcontrollers and adjusted in real-time based on sensor feedback. Furthermore, transfer learning could enable a generalized RL model to be fine-tuned to specific café environments or customer taste profiles using limited additional data. Cloud-based systems could also allow centralized policy updates, aggregating feedback across multiple locations. Integration with IoT sensors for temperature, flow, and mass would further support real-time monitoring, while a user interface could enable customers to influence learning preferences (*e.g.*, strength, acidity). These implementations position RL as a promising method for scalable, intelligent beverage control in commercial settings.

This study is limited to a few extraction variables: grind size, brew ratio, brew time, and temperature. Future research could include additional variables such as flow rate, the number of pourings, and other factors critical to the pour-over coffee process. Beyond pour-over coffee, future studies might also investigate the application of RL to optimize extraction variables in other coffee brewing techniques, such as machine-based methods. Furthermore, RL's potential extends beyond coffee-related research and could contribute to other dynamic and complex domains. Another limitation of this study is the modest sample size and reliance on convenience sampling. While this approach provided practical insights from skilled practitioners, future work should aim to diversify and enlarge the dataset by involving participants from multiple locations, skill levels, and coffee varieties to enhance the robustness and generalizability of the findings.

Additionally, the current approach presents limitations related to potential overfitting and computational efficiency. Since the reinforcement learning agent is trained on a limited set of simulated episodes tailored to specific brewing variables, there is a risk of

overfitting to the training environment, which may reduce generalizability to real-world or more diverse coffee scenarios. Moreover, the learning process requires substantial computational resources and their latency, particularly during simulation-based training across thousands of episodes. This may constrain its use in embedded systems or low-power brewing devices without further model optimization or transfer learning techniques.

Although this study demonstrates the feasibility of applying RL to optimize pour-over coffee extraction variables, it does not yet include benchmarking against current state-of-the-art RL architectures. Recent models such as Proximal Policy Optimization (*Schulman et al., 2017*) and Soft Actor-Critic (*Haarnoja et al., 2018*) have shown strong performance in continuous control tasks, owing to improved stability and sample efficiency. While our implementation used a discrete RL setup suitable for our constrained simulation, future work should incorporate these more advanced frameworks to benchmark policy generalization and extraction performance.

One key consideration in deploying AI for beverage optimization is explainability. Reinforcement learning agents, particularly those trained with deep Q-networks or similar function approximators, can often act as black boxes, making it difficult to trace their decision-making logic. In this study, we provided some insight by analyzing the frequency and consistency of variable-action pairs across episodes. However, future work should explore more advanced explainability techniques. These include SHapley Additive exPlanations (SHAP), which assigns feature importance values for model predictions (*Lundberg & Lee, 2017*), attention-based mechanisms to visualize which inputs guide the agent's policy decisions (*Wang, Lian & Yu, 2021*), and saliency mapping methods to highlight influential state variables in deep reinforcement learning models (*Zheng et al., 2021*). Incorporating such tools would enhance transparency and trust, particularly in applications like personalized or commercial coffee brewing systems where human oversight is essential.

This study also raises important ethical and data bias considerations. The use of convenience sampling from a specific location and demographic (*i.e.*, experienced baristas in a single café) may limit the representativeness of brewing preferences and practices across cultures or consumer segments. Additionally, the reliance on a simulated environment may inadvertently encode assumptions that do not account for diverse sensory expectations. Finally, while reinforcement learning can support consistency and automation in coffee brewing, its use should not undervalue the role of human creativity and judgment in specialty coffee practices. Future studies should consider incorporating diverse data sources and participatory design approaches to ensure inclusive, transparent, and responsible AI applications in this space.

Recent advancements in RL applications within food science further support the relevance of our approach. For instance, a 2024 study by *Queiroz et al. (2023)* demonstrated the use of RL in flavor engineering, developing a framework to discover natural flavor molecules, which highlights RL's potential in optimizing sensory attributes in food and beverage contexts. This aligns closely with our goal of optimizing coffee brewing parameters for improved flavor consistency. Similarly, *Wohlgenannt et al. (2024)*

applied deep RL to optimize energy demand response in a food-processing plant, achieving significant cost savings, which demonstrates RL's broader utility in food industry optimization tasks, such as resource management, that can indirectly enhance beverage production processes. While reinforcement learning has been applied in some areas of beverage and food optimization (*Queiroz et al., 2023*; *Wohlgenannt et al., 2024*), these studies did not address the manual, real-time adjustment of extraction variables in craft brewing methods. To the best of our knowledge, this is the first study to apply RL for optimizing interdependent variables such as grind size, brew ratio, time, and temperature in the pour-over coffee context.

Recent advancements in federated learning (FL) present promising avenues for decentralized data analysis in the coffee industry. FL allows multiple stakeholders, such as coffee producers and processors, to collaboratively train machine learning models without sharing sensitive data, thereby preserving privacy and complying with data protection regulations. This decentralized approach is particularly beneficial in the coffee sector, where data is often distributed across various entities with varying data quality and standards. Moreover, integrating FL with blockchain technology can enhance transparency and fairness in the coffee supply chain, addressing issues related to data integrity and trust among stakeholders. Future research should explore the implementation of FL frameworks tailored to the unique challenges of the coffee industry, including data heterogeneity and the need for equitable data sharing mechanisms.

## CONCLUSIONS

The application of RL for optimizing pour-over coffee extraction variables enables both novice and beginner baristas to identify and achieve optimal extraction conditions. Additionally, this method improves the extraction quality of suboptimal brews. Given the dynamic and interdependent nature of the pour-over coffee extraction process, RL facilitates exploration across various brewing variables or recipes, reducing coffee waste. The RL agent also enhances the effectiveness and efficiency of baristas by providing a tool for experimenting with and controlling brewing parameters while serving as a partner for comparing and sharing brewing recipes.

This study demonstrated that the RL model outperformed the baseline supervised learning model, such as KNN, in optimizing pour-over coffee variables. The RL model achieved an accuracy of 90.00%, a precision of 90.76%, a recall of 90.00%, and an F1-score of 90.08%. In contrast, the KNN model showed slightly lower performance, with an accuracy of 88.33%, a precision of 90.19%, a recall of 88.33%, and an F1-score of 88.90%. The superior performance of the RL model is attributed to its ability to combine exploration, where random actions are taken, and exploitation, which leverages past experiences. Moreover, the RL agent's ability to effectively process reward signals resulted in higher average rewards and scores compared to KNN. Specifically, the RL model achieved an average reward of 3.87 and an average score of 146.93, compared to KNN's average reward of 3.80 and score of 119.87.

Compared to other baseline models—KNN, decision tree, and SVM—the RL framework demonstrated clear superiority in optimizing pour-over coffee brewing

variables, achieving significantly higher evaluation metrics and better consistency across test episodes. While the results validated the approach through extraction yield and reward maximization, the broader implications extend beyond the experimental setup. By demonstrating that RL agents can autonomously learn high-quality brewing strategies, this work lays the groundwork for integrating AI into artisanal processes, helping to balance consistency and personalization. Such systems could enhance user experiences in smart home appliances, support training in barista education, and reduce waste in high-end coffee production. Furthermore, this approach offers a model for human-AI collaboration in food preparation, where transparent, adaptive, and scalable systems can bridge traditional craftsmanship with modern automation.

In future work, additional brewing variables could be incorporated using technically defined pathways. For example, flow rate could be modeled using a digital scale and flow sensor to record real-time mass change and derive precise pouring profiles. Pour intervals and agitation patterns could be encoded using discrete event timestamps or motion sensors integrated into brewing hardware. These new inputs would require extending the RL environment with multi-dimensional state representations and reward functions calibrated using sensory metrics or expert feedback. Furthermore, a hybrid human-in-the-loop setup could be tested, where the RL agent makes suggestions and human baristas provide corrective feedback, accelerating convergence and interpretability.

### Funding
This research work was funded by Institutional Fund Projects under grant no. (IFPIP: 1176-611-1443). The authors received technical and financial support from the Ministry of Education and King Abdulaziz University, Deanship of Scientific Research (DSR), Jeddah, Saudi Arabia. The funders had no role in study design, data collection and analysis, decision to publish, or preparation of the manuscript.

### Grant Disclosures
The following grant information was disclosed by the authors:
Institutional Fund Projects: IFPIP: 1176-611-1443.
Ministry of Education and King Abdulaziz University, Deanship of Scientific Research (DSR), Jeddah, Saudi Arabia.

### Competing Interests
The authors declare that they have no competing interests.

### Author Contributions
- Arif Bramantoro conceived and designed the experiments, performed the experiments, analyzed the data, performed the computation work, prepared figures and/or tables, authored or reviewed drafts of the article, and approved the final draft.

- Moch Riyadi Maskur A conceived and designed the experiments, performed the experiments, performed the computation work, prepared figures and/or tables, and approved the final draft.
- Ahmad A. Alzahrani performed the experiments, analyzed the data, authored or reviewed drafts of the article, and approved the final draft.
- Zhahirah Jeffery conceived and designed the experiments, performed the experiments, analyzed the data, prepared figures and/or tables, authored or reviewed drafts of the article, and approved the final draft.
- Ary Mazharuddin Shiddiqi analyzed the data, performed the computation work, authored or reviewed drafts of the article, and approved the final draft.

## Data Availability

The data is available at figshare: Bramantoro, Arif (2025). Optimization of pour-over coffee extraction. figshare. Dataset. https://doi.org/10.6084/m9.figshare.28503464.v1.

## Supplemental Information

Supplemental information for this article can be found online at http://dx.doi.org/10.7717/peerj-cs.3219#supplemental-information.

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
