# Peer review of "Optimization of pour-over coffee extraction variables using reinforcement learning"

_PeerJ Computer Science, doi:10.7717/peerj-cs.3219_

## Round 0.1 · original submission · Major Revisions

Please revise the manuscript in line with the reviewer comments.

**Language Note:** The review process has identified that the English language must be improved. PeerJ can provide language editing services - please contact us at [email protected] for pricing (be sure to provide your manuscript number and title). Alternatively, you should make your own arrangements to improve the language quality and provide details in your response letter. – PeerJ Staff

Reviewer 1 ·

Basic reporting

- Typo: "cofee" instead of Coffee in keywords.
- TDS (Total Dissolved Solids) is undefined in the abstract/intro.
- Overlap between the abstract and introduction.
- Lacks 2024 references for RL in food science (e.g., recent RL applications in beverage optimization).
- Insufficient justification for RL vs. other ML methods.
- Relies on pre-2023 studies for RL fundamentals (Sutton & Barto, 2018).
- Limited discussion of 2024 advancements in dynamic optimization. - Abbreviations (e.g., TDS, SCA) are not consistently defined.
- Redundant content in abstract/intro.
- Correct typos and define all abbreviations upfront.
- Add 2024 references (e.g., RL in food chemistry).
- Merge redundant sections and clarify the novelty of the RL application.

Experimental design

- Small sample size (10 data points) with convenience sampling.
- Hyperparameters (e.g., buffer size=2000) lack justification.
- KNN implementation details are missing.
- No comparison with hybrid or deep RL models (common in 2024 food science studies).
- Limited use of recent RL frameworks (e.g., stable-baselines3 is cited but not benchmarked).
- The methods section omits 2024 best practices (e.g., reproducibility checklists).
- Hardware/software details lack versioning.
- Technical jargon (e.g., MDP, epsilon-greedy) inadequately explained.
- Tables/figures fragmented (e.g., Table 1 split across pages).

It is essential to:

- Justify the sampling method and increase data diversity.
- Provide hyperparameter sensitivity analysis.
- Share code/datasets and specify software versions.
- Simplify RL/KNN explanations with analogies (e.g., "exploration vs. exploitation as trial-and-error cooking").

Validity of the findings

- Metrics lack statistical validation (no confidence intervals/p-values).
- No blind taste tests or expert validation of brewed coffee quality.
- Does not address 2024 critiques of RL reward design (e.g., sparse rewards in dynamic systems).
- Misses comparisons with deep RL (e.g., DQN, PPO).
- Results focus on accuracy but omit real-world applicability (e.g., scalability). - Figures (e.g., learning curves) lack descriptive captions.
- Overemphasis on numerical results without qualitative insights.

Suggestions for revision:
- Add statistical tests (e.g., t-tests for RL vs. KNN).
- Conduct sensory evaluations with baristas.
- Discuss limitations (e.g., overfitting risks, computational costs).
- Update discussion with 2024 RL advancements (e.g., reward shaping).

Additional comments

- Novelty overstated without addressing prior RL applications in beverage optimization.
- Limited discussion of ethical/data bias concerns.
- Mixed citation years (e.g., 2023 references in a 2025 manuscript).
- Fails to cite 2024 studies on personalized brewing AI.
- The application is novel but lacks benchmarking against 2024 SOTA models.
- No mention of explainability (critical in 2024 AI research).
- Inconsistent formatting (e.g., figure references in text vs. placement).
- Complex equations lack context for non-experts.

- Future work suggestions are generic (e.g., "add more variables") without technical pathways.
- Does not contextualize RL’s scalability for commercial use.
- Omits 2024 trends (e.g., federated learning for decentralized coffee data).
- Fails to reference 2024 studies on sustainable AI in food systems.
- Limited discussion of RL’s real-world deployment challenges (e.g., latency).
- Conclusions reiterate results without synthesizing broader implications

Reviewer 2 ·

Basic reporting

The 1st paragraph introduces the concept of the three waves in the coffee industry, but does not provide any explanation of what the first and second waves entail. This lack of context makes it difficult for readers to understand why the third wave is particularly emphasized. It would be helpful to briefly define each wave before discussing the significance of the third wave, especially in the context of Indonesia. Consider adding a short explanation of the first and second waves to provide a clearer background.

Line 88
The paragraph introduces the role of machine learning in optimizing the brewing process, but focuses specifically on reinforcement learning (RL) without explaining why RL is chosen over other machine learning techniques. Given the wide range of ML methods available, it would be beneficial to provide additional references that highlight RL’s advantages in this context. Additionally, discussing previous research on method selection for pour-over brewing would strengthen the argument and provide a clearer justification for using RL.
Consider adding a brief comparison of RL with other ML techniques commonly used in coffee preparation.

Line 223
The abstract mentions that the study will compare the proposed method with K-Nearest Neighbors (KNN), but the introduction does not provide any discussion on KNN or the rationale for choosing it as the benchmark method. To improve clarity and coherence, it would be beneficial to briefly explain KNN in the introduction and justify why it was selected for comparison instead of other machine learning techniques.

Consider adding a short discussion in the introduction on KNN’s relevance to the problem domain.

LIne 246
The equations presented in the paper do not include citations to indicate their source. If these equations are derived from previous research or standard formulations, it is important to provide proper references. This not only strengthens the credibility of the paper but also allows readers to trace the origins of the mathematical models used.

Consider adding citations after each equation where applicable.

If the equations are newly formulated in this study, a brief explanation should be provided to clarify how they were derived.

Experimental design

-

Validity of the findings

-

Additional comments

Please check the reference writing.

Annotated reviews are not available for download in order to protect the identity of reviewers who chose to remain anonymous.

---

## Round 0.2 · Minor Revisions

Thank you for your submission and the thoughtful work presented in your manuscript. After careful review, I am pleased to inform you that your paper is recommended for minor revision.

Reviewer 2 noted that while the manuscript is clearly written and contextually grounded, it would benefit from a more explicit comparison between the proposed reinforcement learning (RL) approach and other established methods. To strengthen the paper, please consider adding a performance comparison with either conventional non-RL techniques (e.g., KNN, SVM, decision trees) or other RL-based methods from prior research. This would provide valuable context to better assess the novelty and practical advantage of your method in the pour-over coffee brewing task.

We look forward to your revised submission and are confident that addressing this point will further enhance the clarity and impact of your work.

Reviewer 2 ·

Basic reporting

Intro & background to show context. Literature well referenced & relevant.

Experimental design

-

Validity of the findings

The paper lacks a clear comparison between the accuracy of the proposed reinforcement learning (RL) method and other commonly used techniques, or with RL-based methods developed by previous researchers. Including such a comparison would provide valuable context for evaluating the effectiveness and novelty of the proposed approach.

Suggested Revision:
Please consider presenting a comparative analysis of the model’s accuracy, either with non-RL methods (e.g., KNN, SVM, or decision trees) or with existing RL-based approaches in related studies. This would help demonstrate the performance advantage and justify the use of RL for the pour-over coffee brewing task.

---

## Round 0.3 · accepted · Accept

The paper has undergone two rounds of peer review, and no further comments were received from the reviewers in the second round. Given that all concerns have been addressed satisfactorily, I am pleased to inform you that the manuscript is accepted for publication.

Reviewer 2 ·

Basic reporting

no comment

Experimental design

no comment

Validity of the findings

no comment